# LLM Meets Diffusion: A Hybrid Framework for Crystal Material Generation

**Subhojyoti Khastagir**[1,*]**, Kishalay Das**[1,†,*]**, Pawan Goyal**[1]**, Seung-Cheol Lee**[2]**,**
**Satadeep Bhattacharjee**[2]**, Niloy Ganguly**[1]

[1] Indian Institute of Technology, Kharagpur, India
[2] Indo Korea Science and Technology Center, Bangalore, India
* Equal Contribution

## Abstract

Recent advances in generative modeling have shown significant promise in designing novel periodic crystal structures. Existing approaches typically rely on either large language models (LLMs) or equivariant denoising models, each with complementary strengths: LLMs excel at handling discrete atomic types but often struggle with continuous features such as atomic positions and lattice parameters, while denoising models are effective at modeling continuous variables but encounter difficulties in generating accurate atomic compositions. To bridge this gap, we propose CrysLLMGen, a hybrid framework that integrates an LLM with a diffusion model to leverage their complementary strengths for crystal material generation. During sampling, CrysLLMGen first employs a fine-tuned LLM to produce an intermediate representation of atom types, atomic coordinates, and lattice structure. While retaining the predicted atom types, it passes the atomic coordinates and lattice structure to a pre-trained equivariant diffusion model for refinement. Our framework outperforms state-of-the-art generative models across several benchmark tasks and datasets. Specifically, CrysLLMGen not only achieves a balanced performance in terms of structural and compositional validity but also generates more stable and novel materials compared to LLM-based and denoising-based models Furthermore, CrysLLMGen exhibits strong conditional generation capabilities, effectively producing materials that satisfy user-defined constraints. Code is available at `https://github.com/kdmsit/crysllmgen`

## 1 Introduction

Discovery of novel three-dimensional crystal materials with desired chemical properties remains a fundamental challenge in the field of materials design. These materials play a critical role in driving innovations such as development of batteries, solar cells, and semiconductors [1, 2]. Generating crystal materials presents a significant challenge due to its intrinsic complex structure. Unlike molecules, which are typically represented as regular graphs, crystal materials are typically modeled by a minimal unit cell containing all the constituent atoms in different coordinates, repeated infinite times in 3D space on a regular lattice, which makes material structures periodic in nature [3]. Hence, generating stable crystal material requires the simultaneous prediction of both discrete (atomic types) and continuous components (atomic coordinates and lattice structures).

Traditionally, material discovery has relied on either computationally expensive density functional theory (DFT) simulations [4, 5, 6] or labor-intensive experimental procedures. However, recent advances in generative modeling have opened up promising directions for designing novel crystal structures with greater efficiency and scalability. Current generative approaches for material generation can be

---

†Correspondence to Kishalay: kishalaydas@kgpian.iitkgp.ac.in

39th Conference on Neural Information Processing Systems (NeurIPS 2025).

| Validity Metric | Denoising Models | | | LLMs | LLM+Diffusion |
| --- | --- | --- | --- | --- | --- |
| | CDVAE [8] | DiffCSP [10] | FlowMM [14] | LLaMA-2(7B) [7] | CrysLLMGen(7B) |
| Structural Validity | 100 | 100 | 96.85 | 96.40 | 99.94 |
| Compositional Validity | 86.70 | 83.25 | 83.19 | 93.30 | 93.55 |

Table 1: Validity Evaluation of Crystal Generation across model classes: Denoising models tend to achieve higher structural validity, while LLMs excel in compositional validity. CrysLLMGen demonstrates a balanced performance, delivering both high structural and compositional validity.

broadly classified into two main categories: (1) autoregressive large language models (LLMs) [7], and (2) denoising-based frameworks, including diffusion models [8, 9, 10, 11, 12, 13] and flow matching techniques [14, 15]. Each of these generative models has distinct advantages and drawbacks. LLMs excel at capturing discrete information, making them particularly effective at predicting atomic types and achieving high compositional validity. However, they often face challenges in accurately generating continuous data such as atomic positions and lattice parameters due to limitations in finite precision encoding, leading to reduced structural validity. In contrast, denoising-based models are inherently better at handling continuous variables and preserving geometric equivariances, resulting in higher structural validity. Nevertheless, these models struggle with discrete components, such as correctly identifying atomic types, which leads to lower compositional validity (Table-1).

In this work, we aim to harness the complementary strengths of both LLMs and denoising models to mitigate their individual limitations. To this end, we introduce CrysLLMGen, a novel hybrid framework for periodic material generation, which consists of two modules: a fine-tuned LLM and a pre-trained Diffusion Model. Our sampling process begins with the LLM that generates intermediate predictions for atom types, coordinates, and lattice. Given that LLMs excel at modeling discrete information, the generated atom types are well-aligned with the true atomic distributions observed in material datasets. Hence, we retain atomic types and then pass the predicted atomic coordinates and lattice structures to a pre-trained equivariant diffusion model, which refines and adjusts these continuous components to produce more stable and structurally valid crystal materials. Our simple yet effective hybrid approach enables the joint modeling of multimodal features, both discrete and continuous, of crystal materials, significantly enhancing the validity, stability, and novelty of the generated structures. A major advantage of incorporating large language models (LLMs) is their ability to process natural language prompts, allowing for conditional generation. Additionally, denoising models handle crystal symmetries and the equivariant nature of material data distributions. Importantly, our proposed framework is architecture-agnostic, meaning it can readily accommodate future advancements in both LLMs and denoising networks without requiring substantial modifications.

Extensive experiments on benchmark datasets demonstrate that hybrid models can effectively capture both the discrete and continuous aspects of crystal structures, resulting in improvements in both compositional and structural validity. Furthermore, our results show that CrysLLMGen generates **32%** and **68%** more stable materials compared to state-of-the-art LLM-based models and best-performing denoising model. To sum up, our novel contributions in this work are as follows:

- To the best of our knowledge, CrysLLMGen is the first hybrid model that combines large language models (LLMs) with diffusion models for crystal material generation.

- Extensive experiments on benchmark datasets demonstrate that CrysLLMGen consistently outperforms state-of-the-art models in both structural and compositional validity. It also generates more stable, unique, and novel crystal structures compared to existing approaches.

- CrysLLMGen shows strong generative capability under conditional prompts, effectively producing materials aligning with specified atomic compositions and space group constraints.

## 2   Related Work: Crystal Material Generation

Earlier works on periodic material generation primarily focused on atomic composition, often overlooking 3D structural details. With the rise of generative models, approaches using VAEs or GANs have aimed to generate 3D periodic structures by representing materials as voxel images [16, 17, 18, 19] or embedding vectors [20, 21, 22]; however, these methods neither ensure structural stability nor maintain invariance to Euclidean or periodic transformations.

**Diffusion Models.** Recently, equivariant diffusion models have become the leading approach for stable crystal material generation due to their ability to utilize the physical symmetries of periodic structures. Models like CDVAE [8] and SyMat [9] combine VAEs with score-based denoising networks operating on atomic coordinates through equivariant GNNs, ensuring Euclidean and periodic invariance. Subsequent models, including DiffCSP [10] and MatterGen [11], jointly learn atomic composition, coordinates, and lattice parameters via diffusion frameworks. UniMat [13] extends this idea with a unified 4D tensor representation that models discrete atom types and continuous coordinates using a probabilistic diffusion model. More recently, text-guided diffusion models such as TGDMat [23] and Chemeleon [24] integrate contextual embeddings from pretrained models like MatSciBERT or CLIP into GNN-based denoising networks to generate stable periodic materials aligned with textual descriptions.

**Symmetry-Aware Generation.** While conventional diffusion models learn atomic positions independently, incorporating space group symmetry greatly reduces model complexity. DiffCSP++ [12] is the first to extend DiffCSP by enforcing symmetry constraints on lattice parameters and atomic coordinates, ensuring correct lattice systems and restricting atoms to Wyckoff positions from training templates. SymmCD [25] further advances this by jointly learning fractional coordinates and Wyckoff positions using a site-symmetry–aware representation. By modeling only one representative atom per crystallographic orbit, both methods substantially reduce generative complexity.

**Latent Diffusion Models.** A major limitation of current diffusion-based methods is their reliance on high-dimensional feature spaces that jointly model atom types, fractional coordinates, and lattice structures: an inherently multimodal distribution with distinct statistical characteristics for each component. This leads to high computational costs during training and inference, restricting their use in resource-limited settings. Recent works such as CrysLDM [26] and ADiT [27] address these challenges through latent diffusion models, which operate in a compact latent space to significantly reduce sampling time and computational overhead, offering improved efficiency over conventional feature-space diffusion approaches.

**Flow matching.** Flow matching (FM) [28, 29, 30] has recently emerged as a strong alternative to diffusion-based methods for crystalline material generation. Unlike diffusion models that iteratively denoise samples from a Gaussian prior, FM directly learns a time-dependent velocity field that continuously transports an arbitrary base distribution toward the target distribution of stable crystals. The first such application, FlowMM [14], introduced a representation that preserves global rotational and translational symmetries and enforces periodic boundary conditions through equivariant flows to ensure symmetry invariance. Building on this, FlowLLM [15] combines geometric inductive biases with base distributions guided by large language models [7]. Additionally, CrysBFN [31] employs periodic Bayesian Flow Networks with entropy conditioning and non-monotonic dynamics to better capture periodicity in non-Euclidean space, enhancing both sampling efficiency and generation quality.

**Large Language Models.** Large language models (LLMs) are increasingly applied in the natural sciences as versatile priors for reasoning over sequences, graphs, and spatial data, a trend that has recently extended to materials generation. By representing crystal structures as textual descriptions of unit cells and atomic positions, token-based language models [32, 7] have shown strong performance in generating stable and valid materials.

**Key Differences with FlowLLM [15].** By design, our proposed CrysLLMGen comes close to FlowLLM [15], however here are key distinctions in design and methodology. Our approach differs from FlowLLM in three key aspects. (1) *Generative Module:* While FlowLLM employs a flow-matching framework as its generative component, our method, CrysLLMGen, utilizes a diffusion-based model. Through comparative analysis on unconditional material generation (Table-2), we observed that the diffusion-based DiffCSP consistently outperforms the flow-matching model FlowMM across most benchmark metrics. (2) *Parallel Training:* Unlike FlowLLM's sequential training paradigm, where a large language model (LLaMA-2) is first fine-tuned on the dataset and its generated samples are subsequently used to train the flow-matching model, our approach trains both the LLM and the diffusion module *in parallel* on the same training dataset. (3) *Integration Strategy:* In FlowLLM, the material structures generated by the LLM are directly refined by the flow-matching module. In contrast, we treat the LLM-generated material structures as *intermediate representations*, which are injected into our diffusion model at an intermediate timestep $\tau$ (where $0 \leq \tau \leq T$) to

initiate denoising from that point. This integration strategy allows more effective refinement and generation, harnessing the complementary strengths of both the LLM and the diffusion model.

# 3 Preliminaries

## 3.1 Crystal Structure Representation

The crystal structure can be visualized as a three-dimensional point cloud, where atoms are positioned at specific coordinates within a minimal unit cell that repeats periodically in space to form the infinite, periodic crystal lattice. Given a material with $N$ number of atoms in its unit cell, we can describe the unit cell by two matrices: *Atom Type Matrix (A)* and *Coordinate Matrix (X)*. Atom Type Matrix $A = [a_1, a_2, ..., a_N]^T \in \mathbb{R}^{N \times k}$ denotes set of atomic type in one hot representation (k: maximum possible atom types). On the other hand, Coordinate Matrix $X = [x_1, x_2, ..., x_N]^T \in \mathbb{R}^{N \times 3}$ denotes atomic coordinate positions, where $x_i \in \mathbb{R}^3$ corresponds to coordinates of $i^{th}$ atom in the unit cell. Further, there is an additional *Lattice Matrix* $L = [l_1, l_2, l_3]^T \in \mathbb{R}^{3 \times 3}$, which describes how a unit cell repeats itself in the 3D space towards $l_1, l_2$ and $l_3$ direction to form the periodic 3D structure of the material. Formally, a given material can be defined as $M = (A, X, L)$ and we can represent its infinite periodic structure as $\hat{X} = \{\hat{x}_i | \hat{x}_i = x_i + \sum_{j=1}^3 k_j l_j\}; \hat{A} = \{\hat{a}_i | \hat{a}_i = a_i\}$ where $k_1, k_2, k_3, i \in Z, 1 \le i \le N$.

## 3.2 Symmetry in Crystal Structure

Crystal structures inherently exhibit a range of physical symmetries, which are fundamental to their characterization and physical properties. Consequently, a major challenge for any generative model designed for crystal generation is to ensure that the learned distribution satisfies periodic E(3) invariance, meaning invariance to permutation, translation, rotation, and periodic transformations. Permutation invariance implies that reordering the indices of the constituent atoms does not alter the identity of the material, whereas translation invariance means that shifting all atomic coordinates by a constant vector leaves the material structure unchanged. Rotational invariance indicates that rotating both the atomic coordinates and the lattice matrix does not affect the material's identity. Periodic invariance arises from the fact that atoms in a unit cell repeat infinitely along the lattice vectors, allowing multiple valid representations through different choices of unit cells and coordinate matrices for the same material.

# 4 Methodology

## 4.1 Problem Formulation

In this work, we consider generative modeling of 3D crystal structures from scratch, to discover new stable materials. Formally, given a dataset $\mathcal{M} = \{M_i\}_{i=1}^m$, containing crystal structure $M_i = (A_i, X_i, L_i)$, the goal is to capture the underlying data distribution $p(M)$ via learning a generative model $f_\theta(M)$, where $\theta$ is a set of learnable parameters. While training, we need $f_\theta$ to ensure that the learned distribution is invariant to different symmetry transformations mentioned in Section 3.2. Once trained, the generative model can sample valid and stable material structures that remain invariant under various symmetry transformations (unconditional generation). Additionally, by providing specific constraints during sampling, the model can generate structures that meet desired criteria (conditional generation).

## 4.2 Proposed Framework: CrysLLMGen

Our proposed framework, CrysLLMGen, is a hybrid model that combines a large language model (LLM) with a diffusion model (DM) for crystal material generation. We fine-tune the LLM and train the diffusion model independently as standalone components using the training dataset. The sampling process begins by prompting the LLM to generate initial predictions for $A$, $X$, and $L$. Given the LLM's strong capability in modeling discrete information, we assume that the predicted atom types closely align with the true atomic distributions found in material datasets. Therefore, we retain these predicted atom types as the final atomic composition of the generated material. The atomic coordinates and lattice parameters are then passed to a diffusion model, which refines these

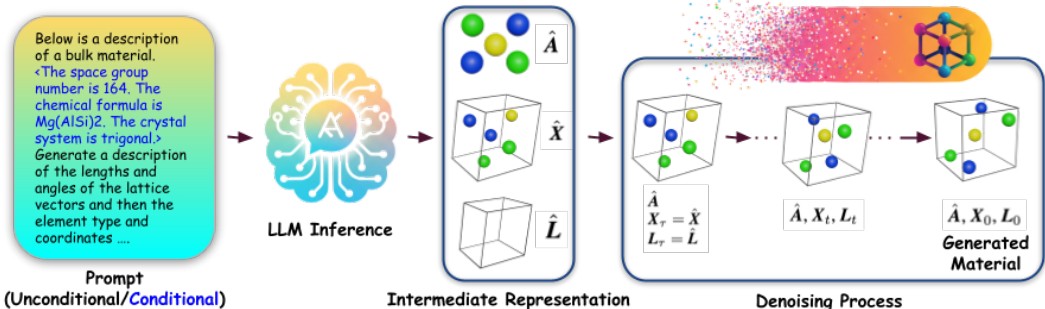

Figure 1: Model Architecture of our proposed CrysLLMGen. The sampling process begins with prompting the LLM to generate intermediate representations for $A$, $X$, and $L$. The predicted atom types ($A$) are retained as-is, while the atomic coordinates ($X$) and lattice parameters ($L$) are further refined through a denoising diffusion network.

continuous components to ensure structural validity and enhance the stability of the generated crystal materials. Next, we will explain in details both the components of our proposed framework.

## 4.3 Large Language Model (LLM) $f_\phi^{LLM}$ for Crystal Materials

We build upon prior work [7] to develop our LLM-based component, in which crystal structures are converted into sequential representations and fed into the LLM for further processing. Fundamentally, LLMs model a distribution over sequences using autoregressive next-token prediction task, where each token is generated based on a categorical distribution conditioned on all preceding tokens in the input sequence. We transform a dataset of crystal structures $\mathcal{M} = \{\boldsymbol{M}_i\}_{i=1}^m$ into a corresponding dataset of sequences $\mathcal{W} = \{W_1, W_2, \ldots, W_m\}$, where each sequence $W_i$ represents the CIF text format of the crystal structure $\boldsymbol{M}_i$. Next, we tokenize each text sequence representing a crystal structure. We represent each crystal material $\boldsymbol{M}_i$ using fixed-precision values: fractional 3D coordinates are rounded to two decimal places, lattice lengths to one decimal place, and angles are encoded as integers. Atom types are treated as discrete tokens. For our LLM backbone, we adopt the state-of-the-art LLaMA-2 models, which have demonstrated strong performance in material generation tasks across several recent studies. Additionally, LLaMA-2 tokenizes numbers as individual digits by default, a feature shown to significantly enhance performance on arithmetic-related tasks.

We use the pre-trained LLaMA-2-7B base model as our starting point. It is fine-tuned on a dataset of crystal structures represented as text sequences. We use task-specific prompts to fine-tune the model on different tasks like unconditional or text-conditional generation. Although larger LLaMA variants such as LLaMA-2-13B, LLaMA-2-70B, or the newer LLaMA-3 models may offer enhanced performance, we opted for the LLaMA-2-7B model to balance computational efficiency, the scale of available pretraining data, and practical deployment considerations for the broader materials science community. Exploring higher-end models is left as a direction for future work.

Crystal structures exhibit translational and rotational symmetries, which standard LLM architectures are not inherently equipped to capture. To address this, we adopt the data augmentation strategy proposed by [7]. Once trained, the LLM can generate sequences by sampling tokens sequentially from the learned categorical distribution.

## 4.4 Diffusion Model $f_\theta^{Diff}$ for Crystal Materials [Atom Coordinates, Lattice Structures]

Prior works leverage equivariant diffusion models to jointly learn atom types $A$, atomic fractional coordinates $X$, and lattice structure $L$. However, in our setup, we retain the atom types predicted by the LLM as the final atomic composition of the generated material, while the atomic coordinates and lattice structure are refined using a diffusion model. To enable this, we train the diffusion model on a structure prediction task, where, given the atom types $A$ of a crystal material, it learns the joint distribution of atomic fractional coordinates $X$ and lattice structure $L$.

**Diffusion on Atom Coordinates ($X$).** Atom coordinates can be diffused in two ways: by applying noise to either the cartesian coordinates or the fractional coordinates. Prior works such as

CDVAE [8] adopt cartesian coordinate diffusion, while DiffCSP [10] operates on fractional coordinates. In our approach, since we jointly learn atomic positions and the lattice matrix, we follow the methodology of DiffCSP and perform diffusion on the fractional coordinates. Coordinate Matrix $X = [x_1, x_2, ..., x_N]^T \in \mathbb{R}^{N \times 3}$ contains fractional coordinates of constituent atoms, that resides in quotient space $\mathbb{R}^{N \times 3}/\mathbb{Z}^{N \times 3}$ induced by the crystal periodicity. Since the Gaussian distribution used in DDPM is unable to model the cyclical and bounded domain of $X$, it is not suitable to apply DDPM to model $X$. Hence at each step of forward diffusion, we add noise sample from Wrapped Normal (WN) distribution [33] to $X$ and during backward diffusion leverage Score Matching Diffusion Networks [34, 35] to model underlying transition probability $q(X_t \mid X_0) = \mathcal{N}_W(X_t \mid X_0, \sigma_t^2 I)$. In specific, at each $t^{th}$ step of diffusion, we derive $X_t$ as : $X_t = f_w(X_0 + \sigma_t \epsilon^X)$ where, $\epsilon^X$ is a noise, sampled from $\mathcal{N}(0, I)$, $\sigma_t$ is the noise scale following exponential scheduler and $f_w(.)$ is a truncation function. Given a fractional coordinate matrix X, truncation function $f_w(X) = (X - \lfloor X \rfloor)$ returns the fractional part of each element of $X$. As argued in [10], $q(X_t|X_0)$ is periodic translation equivariant, and approaches uniform distribution $\mathcal{U}(0, 1)$ for sufficiently large values of $\sigma_T$. We use a denoising network $\Phi_\theta(A_t, X_t, L_t, t)$ to model the backward diffusion process, which is trained using the following score-matching objective function :

$$\mathcal{L}_{coord} = \mathbb{E}_{\substack{X_t \sim q(X_t|X_0) \\ t \sim \mathcal{U}(1,T)}} \|\nabla_{X_t} \log q(X_t|X_0) - \hat{\epsilon}^X(M_t, t)\|_2^2 \tag{1}$$

where $\nabla_{X_t} \log q(X_t|X_0) \propto \sum_{K \in \mathbb{Z}^{N \times 3}} \exp(- \frac{\|X_t - X_0 + K\|_F^2}{2\sigma_t^2})$ is the score function of transitional distribution and $\hat{\epsilon}^X(M_t, t)$ denoising term.

**Diffusion on Lattice Structure ($L$).** Lattice Matrix $L = [l_1, l_2, l_3]^T \in \mathbb{R}^{3 \times 3}$ is a global feature of the material which determines the shape and symmetry of the unit cell structure. Since $L$ is in continuous space, we leverage the idea of the Denoising Diffusion Probabilistic Model (DDPM) [36] for diffusion on $L$. Specifically, given an initial lattice matrix $L_0 \sim p(L)$, the forward diffusion process gradually corrupts it over $T$ timesteps, resulting in a noisy lattice matrix $L_T$. At each timestep $t$, the transition is governed by a conditional probability distribution $q(L_t \mid L_0)$, which can be formally expressed as: $q(L_t \mid L_0) = \mathcal{N}(L_t \mid \sqrt{\bar{\alpha}_t} L_0, (1 - \bar{\alpha}_t) I)$, where, $\bar{\alpha}_t = \prod_{k=1}^t \alpha_k$, $\alpha_t = 1 - \beta_t$ and $\{\beta_t \in (0,1)\}_{t=1}^T$ controlling the level of noise added at each step. By reparameterization, we can rewrite as: $L_t = \sqrt{\bar{\alpha}_t} L_0 + \sqrt{1 - \bar{\alpha}_t} \epsilon^L$ where, $\epsilon^l$ is a noise, sampled from $\mathcal{N}(0, I)$, added with original input sample $L_0$ at $t^{th}$ step to generate $L_t$. After T such diffusion steps, noisy lattice matrix $L_T$ is generated from prior noise distribution $\sim \mathcal{N}(0, I)$. During denoising, the reverse conditional distribution can be expressed as follows : $p(L_{t-1}|M_t) = \mathcal{N}\{L_{t-1} \mid \mu^L(M_t), \beta_t \frac{(1-\bar{\alpha}_{t-1})}{(1-\bar{\alpha}_t)} I\}$, where $\mu^L(M_t) = \frac{1}{\sqrt{\alpha_t}}(L_t - \frac{1-\alpha_t}{\sqrt{1-\bar{\alpha}_t}} \hat{\epsilon}^L(M_t, t))$. Intuitively, $\hat{\epsilon}^l$ is the denoising term that needs to be subtracted from $L_t$ to generate $L_{t-1}$. We use a denoising network $\Phi_\theta(A_t, X_t, L_t, t)$ to model the noise term $\hat{\epsilon}^L(M_t, t)$. Following the simplified training objective proposed by [36], we train the aforementioned denoising network using $l_2$ loss between $\hat{\epsilon}^L$ and $\epsilon^L$

$$\mathcal{L}_{lattice} = \mathbb{E}_{\epsilon^L, t \sim \mathcal{U}(1,T)} \|\epsilon^L - \hat{\epsilon}^L\|_2^2 \tag{2}$$

**Denoising Network.** For the denoising network $f_\theta(A_t, X_t, L_t, t)$ in the reverse diffusion process, we extend the CSPNet architecture [10], which is built upon the Equivariant Graph Neural Network (EGNN) [37]. This architecture is specifically designed to satisfy the periodic E(3) invariance conditions inherent in periodic crystal structures. At the $k^{th}$ layer message passing, the Equivariant Graph Convolutional Layer (EGCL) takes as input the set of atom embeddings $h^k = [h_1^k, h_2^k, ..., h_N^k]$, atom coordinates $x^k = [x_1^k, x_2^k, ..., x_N^k]$ and Lattice Matrix $L$ and outputs a transformation on $h^{k+1}$. Formally, we can define the $k^{th}$ layer message passing operation as follows :

$$m_{i,j} = \rho_m\{h_i^k, h_j^k, L^T L, \psi_{FT}(x_i^k - x_j^k)\}; \quad m_i = \sum_{j=1}^N m_{i,j}; \quad h_i^{k+1} = h_i^k + \rho_h\{h_i^k, m_i\} \tag{3}$$

where, $\rho_m$ and $\rho_h$ are multi-layer perceptrons (MLPs), and $\psi_{FT}$ denotes a Fourier Transformation function applied to the relative difference between fractional coordinates $x_i^k$ and $x_j^k$. The use of Fourier Transformation ensures invariance to periodic translations and captures a spectrum of frequencies from the relative fractional distances, which is beneficial for modeling the periodic nature of crystal structures. Input atom features $h^0$ and coordinates $x^0$ are fed through $\mathcal{K}$ layers of EGCL to produce

$\hat{\epsilon}^L$ and $\hat{\epsilon}^X$ as follows :

$$\hat{\epsilon}^L = \boldsymbol{L}\rho_L(\frac{1}{N}\sum_{N}^{i=1}\boldsymbol{h}^{\mathcal{K}}); \;\; \hat{\epsilon}^X = \rho_X(\boldsymbol{h}^{\mathcal{K}}) \tag{4}$$

where $\rho_L, \rho_X$ are multi-layer perceptrons on the final layer embeddings.

---

**Algorithm 1** Sampling Process of CrysLLMGen

---

1: **Input:** A pretrained LLM $f_\phi^{LLM}$, A pretrained Diffusion Model $f_\theta^{Diff}$, (Conditional/Unconditional) Prompt for Material $\mathcal{P}$, Intermediate step $\tau$

2: **Step 1: Sample from LLM $f_\phi^{LLM}$**

3: Sample $\hat{\boldsymbol{A}}, \hat{\boldsymbol{X}}$ and $\hat{\boldsymbol{L}}$ from LLM given prompt $\mathcal{P}$ : $(\hat{\boldsymbol{A}}, \hat{\boldsymbol{X}}, \hat{\boldsymbol{L}}) \sim f_\phi^{LLM}(\mathcal{P})$

4: **Step 2: Refinement using Diffusion Models $f_\theta^{Diff}$**

5: $\boldsymbol{A}_0 := \hat{\boldsymbol{A}}/*Retain*/$

6: $\boldsymbol{L}_\tau := \hat{\boldsymbol{L}}; \boldsymbol{X}_\tau := \hat{\boldsymbol{X}}$

7: **for** $t \leftarrow \tau$ to 1 **do**

8: $\quad \hat{\epsilon}^X, \hat{\epsilon}^L \leftarrow f_\theta(\hat{\boldsymbol{A}}, \boldsymbol{X}_t, \boldsymbol{L}_t, t)$

9: $\quad \boldsymbol{L}_{t-1} \leftarrow \frac{1}{\sqrt{\alpha_t}}(\boldsymbol{L}_t - \frac{\beta_t}{\sqrt{1-\bar{\alpha}_t}}\hat{\epsilon}^L) + \sqrt{\beta_t\frac{1-\bar{\alpha}_{t-1}}{1-\bar{\alpha}_t}}\epsilon; \; \epsilon \sim N(0, I)$

10: $\quad \boldsymbol{X}_{t-\frac{1}{2}} \leftarrow w(\boldsymbol{X}_t + (\sigma_t^2 - \sigma_{t-1}^2)\hat{\epsilon}^X + \frac{\sigma_{t-1}\sqrt{\sigma_t^2-\sigma_{t-1}^2}}{\sigma_t}\epsilon; \; \epsilon \sim N(0, I)$

11: $\quad \_, \hat{\epsilon}^X \leftarrow f_\theta(\hat{\boldsymbol{A}}, \boldsymbol{X}_{t-\frac{1}{2}}, \boldsymbol{L}_{t-1}, t)$

12: $\quad \eta_t \leftarrow step\_size * \frac{\sigma_{t-1}}{\sigma_t}$

13: $\quad \boldsymbol{X}_{t-1} \leftarrow w(\boldsymbol{X}_{t-\frac{1}{2}} + \eta_t\hat{\epsilon}^X + \sqrt{2\eta_t}\epsilon^X)$

14: **end for**

15: **Output:** New Generated Crystal Material: $\boldsymbol{M}_{new} = (\hat{\boldsymbol{A}}, \boldsymbol{X}_0, \boldsymbol{L}_0)$

---

## 4.5 Training and Sampling

We train both components of our proposed framework, CrysLLMGen, the LLM and the diffusion model, independently using the training dataset. For the LLM component, we start with the pretrained LLaMA-2-7B base model and fine-tune it with the Low-Rank Adapters (LoRA) method on crystal structures represented as text sequences, guided by task-specific prompts for either conditional or unconditional generation. In this work, we adopt the prompt formats proposed in [7] for both tasks. The diffusion model is trained for a structure prediction task using a combined loss function: $\mathcal{L} = \mathcal{L}_{lattice} + \mathcal{L}_{coord}$.

Once trained, the sampling process begins with the LLM component. A task-specific prompt is provided as input, and the LLM generates a sequence by sampling tokens sequentially from its learned distribution.[1] This yields an intermediate representation consisting of predicted atom types $\hat{\boldsymbol{A}}$, atomic coordinates $\hat{\boldsymbol{X}}$, and lattice structure $\hat{\boldsymbol{L}}$. Given the LLM's strong capacity for modeling discrete information, we hypothesize that $\hat{\boldsymbol{A}}$ closely aligns with the true atomic distributions observed in material datasets. Therefore, we retain $\hat{\boldsymbol{A}}$ as the final atomic composition of the generated material. But, the continuous variables $\hat{\boldsymbol{X}}$ and $\hat{\boldsymbol{L}}$ require further refinement and hence we need to feed them to the diffusion model. Typically, diffusion models begin the sampling process with fully noisy inputs, i.e., $\boldsymbol{X}_T, \boldsymbol{L}_T \sim \mathcal{N}(\boldsymbol{0}, \mathbf{I})$ and iteratively denoise them over $T$ steps to generate $\boldsymbol{X}_0$ and $\boldsymbol{L}_0$, approximating the target data distribution. However, in our case, $\hat{\boldsymbol{X}}$ and $\hat{\boldsymbol{L}}$ produced by the LLM are not pure noise but meaningful intermediate representations. Therefore, instead of starting the denoising process at step $T$, we inject these representations at an intermediate timestep $\tau$, where

---

[1]LLMs sometimes hallucinate and generate invalid or unphysical chemical elements. To address this, we employ a simple validation strategy that checks and filters compositions during sampling, immediately discarding any invalid ones. On average, about 2–5% of the sampled structures are removed due to invalid atom types or compositions.

| Dataset | Category | Model | Validity | | Coverage | | Property | |
|---|---|---|---|---|---|---|---|---|
| | | | Structural ↑ | Compositional ↑ | Precision ↑ | Recall ↑ | Density ↓ | # Element ↓ |
| MP-20 | Diffusion Models | CDVAE | **100** | 86.70 | 99.49 | 99.15 | 0.687 | 1.432 |
| | | DiffCSP | **100** | 83.25 | 99.76 | 99.71 | 0.352 | 0.339 |
| | | DiffCSP++ | 99.94 | 85.12 | 99.59 | 99.73 | 0.235 | 0.375 |
| | | MatterGen | **100** | 86.34 | 99.45 | 99.59 | 0.459 | 0.254 |
| | | UniMat | 97.20 | 89.40 | 99.70 | **99.80** | **0.088** | **0.056** |
| | | SymmCD | 92.30 | 87.13 | 98.78 | 97.33 | 0.531 | 0.210 |
| | Flow Matching | FlowMM | 96.85 | 83.19 | 99.58 | 99.49 | 0.239 | 0.083 |
| | | FlowLLM | 99.94 | 90.84 | 99.82 | 96.95 | 1.142 | 0.150 |
| | Bayesian Flow Networks | CrysBFN | **100** | 87.51 | 99.79 | 99.09 | 0.206 | 0.163 |
| | LLMs | Lama-2 (7B) | 97.70 | **93.55** | 99.32 | 96.95 | 1.575 | 0.272 |
| | LLM + Diffusion | CrysLLMGen (7B) | 99.94 | **93.55** | **99.84** | 98.52 | 0.972 | 0.272 |
| Perov-5 | Diffusion Models | CDVAE | **100** | 98.59 | 98.46 | 99.45 | 0.126 | 0.063 |
| | | DiffCSP | **100** | 98.85 | 98.27 | **99.74** | 0.111 | 0.013 |
| | | DiffCSP++ | **100** | 98.77 | 98.80 | 99.60 | **0.066** | **0.004** |
| | | MatterGen | 99.84 | 98.21 | 98.17 | 98.96 | 0.109 | 0.036 |
| | | UniMat | **100** | 98.80 | 98.20 | 99.20 | 0.076 | 0.025 |
| | Flow Matching | FlowMM | **100** | 97.91 | 88.91 | 99.31 | 1.210 | 0.061 |
| | | FlowLLM | 99.70 | 98.06 | 90.27 | 99.40 | 0.892 | 0.060 |
| | Bayesian Flow Networks | CrysBFN | **100** | 98.86 | 98.63 | 99.52 | 0.073 | 0.010 |
| | LLMs | Llama-2 (7B) | 99.09 | **98.92** | 98.36 | 98.46 | 0.649 | 0.043 |
| | LLM + Diffusion | CrysLLMGen (7B) | **100** | **98.92** | **98.82** | 99.31 | 0.137 | 0.043 |

Table 2: Summary of results on *De Novo Generation Task* of Different Class of Generative Models.

$0 \leq \tau \leq T$, and initiate the denoising process from there to refine both the coordinates and the lattice structure. $\tau$ is a hyper-parameter, which we choose based on the validation set for each dataset. The final atomic coordinates $X_0$ and lattice structure $L_0$, combined with the retained atom types $\hat{A}$, constitute the generated crystal material, denoted as: $M_{\text{new}} = (\hat{A}, X_0, L_0)$. The full sampling process is described in Algorithm 1.

# 5 Experiments

In this section, we present a comprehensive evaluation of our method against several baselines on three benchmark tasks: De Novo Material Generation (Section 5.1), Stable, Unique, and Novel (S.U.N.) Materials Generation (Section 5.2), and Text-Conditioned Generation (Section 5.3).

## 5.1 De Novo Material Generation (Gen)

**Setup.** First, we focus on *De Novo Material Generation (Gen)*, an unconditional generation task aimed at producing novel, stable crystal materials that are distributionally similar to those in the test dataset. To evaluate the effectiveness of CrysLLMGen in this task, we compare it against seven state-of-the-art generative models across different categories: CDVAE [8], DiffCSP [10], DiffCSP++ [12], MatterGen [11], UniMat [13] and SymmCD [25] from the class of denoising models; FlowMM [14] and FlowLLM [15] from the flow matching paradigm; CrysBFN [31] from the Bayesian Flow Networks and LLaMA-2(7B) [7] representing the LLM-based approach. We use two popular material datasets for this task: Perov-5 [38, 39] and MP-20 [40]. While training all competitive models, we followed the standard dataset split of 60% for training, 20% for validation, and 20% for testing. Following [8], we assess all models using seven evaluation metrics grouped under three broad categories: validity, coverage, and property statistics. (More details in Appendix C) **Results.** We present the results in Table 2. We observe that across both datasets, denoising models such as diffusion and flow matching frameworks excel in structural validity, while LLM models perform better in compositional validity. In contrast, CrysLLMGen, as a unified model, surpasses all baseline models in both structural and compositional validity. Specifically, on MP-20 dataset, CrysLLMGen demonstrates a **4.64%** improvement in compositional validity over leading denoising models and a **2.29%** gain in structural validity over LLM-based models. Furthermore, in coverage metrics, CrysLLMGen achieves the highest performance in COV-Precision, however in COV-Recall, it outperforms LLMs and delivers competitive results against denoising models. Lastly, in terms of property statistics, recent denoising models such as UniMat and FlowMM demonstrate strong performance, while approaches incorporating LLMs (e.g., LLaMA or FlowLLM) tend to underperform. We observe that CrysLLMGen consistently surpasses LLM-based baselines and performs comparably to SOTA denoising models. Overall, CrysLLMGen exhibits promising performance in the material generation task, effectively leveraging the strengths of both LLMs

and diffusion models to enhance generative capability and produce more valid, periodic 3D crystal structures.

## 5.2 Stable, Unique and Novel (S.U.N.) Materials

The ultimate objective of materials discovery is to efficiently screen for stable, unique, and innovative materials. Thermodynamic stability provides a strong indication of a material's synthesizability. A crystal is considered stable when it is energetically more favorable than any alternative structures composed of the same atomic elements, but in different proportions or arrangements.The stability gap can be rationalised by considering the factors contributing to the energy above the convex hull for a given material structure $(A, X, L)$. $E_{\text{hull}}(A, X, L)$ can be conceptually decomposed into contributions related to the intrinsic stability of the composition $A$ itself, and the relative stability of the specific structure $(X, L)$ for that composition,

$$E_{\text{hull}}(\boldsymbol{A}, \boldsymbol{X}, \boldsymbol{L}) \;=\; \underbrace{\Delta E_{\text{struct}}(\boldsymbol{X}, \boldsymbol{L} \mid \boldsymbol{A})}_{\substack{\text{Polymorph} \\ \text{energy split}}} \;+\; \underbrace{\Delta E_{\text{chem}}(\boldsymbol{A})}_{\substack{\text{Compositional} \\ \text{instability}}}. \tag{5}$$

Such stable structures lie directly on or below the convex hull [11]. Once a set of stable materials is obtained after structural relaxation, the next objective is to identify those that are structurally unique and truly novel within that stable set. In this section, we evaluate the effectiveness of our proposed framework in generating stable, unique, and novel materials. To this end, we adopt the S.U.N. (Stability, Uniqueness, and Novelty) metrics introduced in prior work [11] and compare our model's performance against several baseline

| Method | % Meta-Stable ↑ | % M.S.U.N. ↑ | % Stable ↑ | % S.U.N ↑ |
|---|---|---|---|---|
| CDVAE | 23.58 | 21.99 | 3.08 | 2.56 |
| DiffCSP | 35.04 | 32.19 | 7.36 | 5.61 |
| DiffCSP++ | 42.39 | 30.56 | 8.58 | 6.55 |
| FlowMM | 31.64 | 22.46 | 4.76 | 3.06 |
| SymmCD | 40.01 | 31.69 | 9.99 | 6.76 |
| Llama-2 (7B) | 56.60 | 26.66 | 12.67 | 4.84 |
| CrysLLMGen (7B) | **62.02** | **35.94** | **16.79** | **9.21** |

Table 3: Results on S.U.N. metrics for different SOTA models.

approaches. Specifically, we follow the evaluation protocol used in [14, 15], wherein 10,000 candidate structures are generated for each state-of-the-art model and subsequently relaxed using a pretrained CHGNet [41] model to estimate their formation energies. These energies are then compared against a convex hull constructed from the Materials Project database. Relaxed structures are labeled as stable if $E^{\text{hull}} < 0.0$ eV/atom and as metastable if $E^{\text{hull}} < 0.1$ eV/atom. Finally, we assess whether the relaxed stable and metastable structures are also unique and novel, reporting them as % S.U.N. and % M.S.U.N., respectively.

We present the results of all competing baseline models along with CrysLLMGen on the MP-20 dataset in Table 3. Overall, CrysLLMGen demonstrates substantial improvements in generating stable, unique, and novel structures. In specific, compared to the best-performing denoising model, CrysLLMGen achieves relative gains of **46.29%** in metastable rate, **11.65%** in MSUN rate, **68.07%** in stability rate, and **36.27%** in SUN rate. Furthermore, when compared to the LLM-based baseline, the improvements are **9.56%** in metastable rate, **34.80%** in MSUN rate, **32.53%** in stability rate, and **90.29%** in SUN rate. Finally, to further assess how effectively CrysLLMGen generates low-energy structures compared to baseline models, we plotted the histogram of the computed $E^{hull}$ distribution for relaxed structures across different methods, as shown in Figure 2(a). The results indicate that CrysLLMGen produces a greater proportion of low-energy structures than all other baseline models, confirming its ability to generate more stable materials. As mentioned, CrysLLMGen introduces a hybrid approach where the LLM predicts chemically valid compositions $A$, which are then fixed while a diffusion model refines lattice and atomic coordinates. This separation leverages the LLM's chemical prior to prioritize stable compositions, resulting in lower formation energies compared to fully joint diffusion models like DiffCSP.(Check detailed analysis in Appendix E)

## 5.3 Text Conditioned Generation

Next, we evaluate CrysLLMGen on conditional crystal generation task. Following the experimental setup proposed by [7], we extend the idea of text-conditional material generation by providing

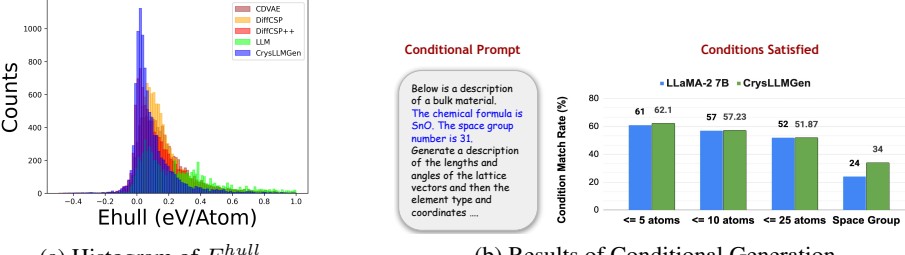

(a) Histogram of $E^{hull}$.           (b) Results of Conditional Generation.

Figure 2: (a) Histogram of $(E^{hull})$ distribution computed after relaxation with CHGNet.(b) Results of Text-Conditional Generation on atomic compositions and space group.

the LLM component with user-defined prompts that specify desired conditions for the generated materials. Specifically, we explore conditioning on atomic composition and space group number by incorporating the target chemical formula and space group into the prompt. This prompt is then passed to the LLM component of our model, which generates materials that aim to satisfy the given constraints. To assess the model's ability to generate crystal materials that satisfy specified atom types and space group constraints, we compare the generated outputs with available ground-truth labels. For chemical composition, we parse the atomic formula from each generated CIF file and check for a match with the target composition. Further, we use Pymatgen's SpacegroupAnalyzer [42] with a tolerance of 0.2 Å to determine the space group of both generated and ground truth materials and compute the match rate. These experiments are conducted on the MP-20 test dataset, and the results are presented in Figure 2(b), where we compare the performance of CrysLLMGen against the LLM baseline proposed by [7]. For atom types, we report match rates across three classes of materials based on the number of atoms: $\leq 5$, $\leq 10$, and $\leq 25$. We observe that CrysLLMGen performs comparably to the LLM baseline across all three categories. Specifically, the model is able to generate materials with the correct composition in most cases, although accuracy declines as the number of atoms in the chemical formula increases. This similar performance is primarily due to CrysLLMGen retaining the atomic compositions provided by the LLM component. However, in the case of space group prediction, CrysLLMGen outperforms the LLM baseline by 42%. This improvement can be attributed to the diffusion model's structural refinement, which enables CrysLLMGen to better align the generated structures with the desired space group.

## 6 Conclusion

In this work, we explore a hybrid approach that integrates large language models (LLMs) with diffusion models for crystal material generation. We propose CrysLLMGen, a two-stage framework that first uses a fine-tuned LLM to generate an intermediate representation of atom types, atomic coordinates, and lattice parameters, and then refines the coordinates and lattice using a pre-trained diffusion model. Extensive experiments on popular material generation tasks demonstrate that CrysLLMGen maintains both structural and compositional validity, outperforming existing baseline models by a good margin. Moreover, CrysLLMGen generates crystal structures that are more stable, unique, and novel than those produced by prior methods. Additionally, it exhibits strong conditional generation capabilities, effectively synthesizing materials that meet user-defined constraints.

## 7 Limitations and Future Work

In our current proposed framework, there is no interaction between the LLM and the diffusion model; both components are trained independently as standalone modules. During sampling, the LLM first generates predictions, which are then passed to the diffusion model for further processing. This separation suggests a potential avenue for future research: exploring whether these two components could benefit from mutual guidance or feedback mechanisms to enhance the overall generation quality. Also, At present, we use a pre-trained LLaMA-2-7B base model as the LLM component, coupled with a vanilla diffusion model that extends DiffCSP. Nonetheless, our framework is designed to be flexible, allowing seamless integration of more advanced LLM variants, such as LLaMA-3, or more fine-grained diffusion models for crystal material generation in future work.

## 8 Acknowledgment

This work was funded by Indo Korea Science and Technology Center, Bangalore, India, under the project name "Generating Stable Periodic Materials using Diffusion Models". We thank the Ministry of Education, Govt. of India, for supporting Kishalay with Prime Minister Research Fellowship (PMRF) during his Ph.D. tenure.

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

# A Technical Appendices and Supplementary Material

# B Related Work(In Details)

Deep learning models are increasingly being applied across various downstream tasks in materials science, including scalable materials design [43], property prediction [44], knowledge-base construction [45], database generation [46], and entity extraction [47]. In this work, we present a comprehensive survey of state-of-the-art methods for crystal material representation learning and generation.

## B.1 Crystal Representation Learning

In recent times, graph neural network (GNN) based approaches have emerged as a powerful model in learning robust representation of crystal materials, which enhance fast and accurate property prediction. CGCNN [48] is the first proposed model, which represents a 3D crystal structure as an undirected weighted multi-edge graph and builds a graph convolution neural network directly on the graph. Following CGCNN, there are a lot of subsequent studies [49, 50, 51, 52, 53, 54], where authors proposed different variants of GNN architectures for effective crystal representation learning. Recently, graph transformer-based architecture Matformer [55] is proposed to learn the periodic graph representation of the material, which marginally improves the performance, however, is much faster than the prior SOTA model. Moreover, scarcity of labeled data makes these models difficult to train for all the properties, and recently, some key studies [56, 57] have shown promising results to mitigate this issue using transfer learning, pre-training, and knowledge distillation respectively.

## B.2 Crystal Material Generation

In the past, there were limited efforts in creating novel periodic materials, with researchers concentrating on generating the atomic composition of periodic materials while largely neglecting the 3D structure. With the advancement of generative models, the majority of the research focuses on using popular generative models like VAEs or GANs to generate 3D periodic structures of materials, however, they either represent materials as three-dimensional voxel images [16, 17, 18, 19] and generate images to depict material structures (atom types, coordinates, and lattices), or they directly encode material structures as embedding vectors [20, 21, 22]. However, these models neither incorporate stability in the generated structure nor are invariant to any euclidean and periodic transformations.

**Diffusion Models.** In recent times equivariant diffusion models have become the leading method for generating stable crystal materials, thanks to their capability to utilize the physical symmetries of periodic material structures. In specific, state-of-the-art models like CDVAE [8] and SyMat [9] integrate a variational autoencoder (VAE) and powerful score-based denoising network, work directly with the atomic coordinates of the structures and uses an equivariant graph neural network to ensure euclidean and periodic invariance. Subsequent models, such as DiffCSP [10] and MatterGen [11], adopt a joint diffusion framework to simultaneously learn atomic composition, fractional coordinates, and lattice parameters. In contrast, UniMat [13] introduces a unified 4D tensor-based crystal representation that jointly models discrete atom types and continuous atomic coordinates using a probabilistic diffusion model with interleaved attention and convolution layers. Lately, there have been a few efforts to develop text-guided diffusion models for text-conditional material generation. Models such as TGDMat [23] and Chemeleon [24] incorporate contextual representations, leveraging pretrained models like MatSciBERT or CLIP, into a GNN-based denoising network. This enables the generation of valid and stable periodic materials that align with the conditions specified in textual descriptions.

**Symmetry-Aware Generation.** While conventional diffusion models independently learn atomic positions within a unit cell, incorporating space group symmetry significantly reduces the model's degrees of freedom. DiffCSP++ [12] is the first approach which builds upon DiffCSP by enforcing space group constraints on both lattice parameters and atomic coordinates. In this approach, lattices are parameterized to ensure the generated structures conform to the correct lattice system, while atomic fractional coordinates are restricted to Wyckoff positions derived from templates in the training data. SymmCD [25] advances this idea by jointly learning the fractional coordinates and corresponding Wyckoff positions of atoms through a site-symmetry–aware representation. By modeling only one representative atom per crystallographic orbit, both SymmCD and DiffCSP++ achieve a substantial reduction in generative complexity.

**Latent Diffusion Models.** A major limitation of current diffusion-based methods lies in their operation within a high-dimensional feature space, where they model the joint distribution of atom types, fractional coordinates, and lattice structures: an inherently multimodal distribution with distinct statistical properties for each component. As a result, these models entail substantial computational cost for both training and inference, limiting their applicability in resource-constrained settings. Few of the recent works like CrysLDM [26], ADiT [27] utilize a latent diffusion model to address the above limitations, reducing sampling time for crystal material generation. Operating in the latent space, these offers unique advantages in generative modeling complexity over existing feature-domain diffusion models, making it more efficient in terms of time and resource consumption.

**Flow matching.** Flow matching (FM) [28, 29, 30] has recently emerged as a promising alternative to diffusion-based methods for crystalline material generation. Unlike diffusion models that iteratively denoise samples from a Gaussian prior, FM directly learns a time-dependent velocity field that continuously transports an arbitrary base distribution toward the target distribution of stable crystals. The first application of this framework, FlowMM [14], introduced a representation that enforces global rotational and translational symmetries along with periodic boundary conditions, while defining base distributions with equivariant flows to ensure symmetry invariance. Building on this, FlowLLM [15] integrates geometric inductive biases with base distributions informed by large language models [7]. Moreover, CrysBFN [31] employs periodic Bayesian Flow Networks (BFNs) with entropy conditioning and non-monotonic dynamics to more effectively capture periodicity in non-euclidean space, thereby improving both sampling efficiency and generation quality.

**Large Language Models.** Large language models (LLMs) are increasingly adapted in the natural sciences as versatile priors for reasoning over sequences, graphs, and spatial data, and this trend has recently extended to materials generation. By encoding crystal structures as textual descriptions of unit cells and atomic positions, token-based language models [32, 7] have demonstrated good performances in generating stable and valid materials.

# C  Experimental Results

## C.1  Benchmark Task

- *De Novo Material Generation (Gen)*: In *Gen* task, the goal of the generative model is to generate novel stable materials (atom types, fractional coordinates, and lattice structure).

- *Conditional Material Generation*: In this task, specific criteria are provided, and the objective is to generate stable crystal materials that satisfy those given conditions.

- *Crystal Structure Prediction (CSP)*: Atom types of the materials are given and the goal is to predict/match the crystal structure (atom coordinates and lattice).

## C.2  Datasets

Following prior works [8] we evaluate our model on three baseline datasets: **Perov-5**, **Carbon-24** and **MP-20**.

- **Perov-5** [38, 39] dataset consists of 18,928 perovskite materials, each with 5 atoms in a cell. They generally can be denoted by $\mathbf{ABX_3}$ indicating the three different types of atoms usually observed in such materials.

- **Carbon-24** [58] dataset has 10,153 materials with 6 to 24 atoms of carbon in the crystal lattice.

- **MP-20** [40] dataset has 45,231 materials curated from the Materials Project library [59], where each material has at most 20 atoms in the lattice.

- **MPTS-52** is a more challenging extension of MP-20, consisting of 40,476 structures up to 52 atoms per cell, sorted according to the earliest published year in literature.

Crystals from **Perov-5** dataset share the same structure but differ in composition, whereas Crystals from **Carbon-24** share the same composition but differ in structure. Crystals from **MP-20** and **MPTS-52** differs in both structure and composition.

### C.3 Hyper-parameter Details

**LLM Component :** We finetune the LLaMA-2 7B model for 1 epoch using the AdamW optimizer implemented via the 'transformers.Trainer' interface. The learning rate is set to 0.0001.

**Diffusion Model :** For the diffusion component, we use a batch size of 256 and adopt a cosine noise schedule. The model is trained for 1000 diffusion steps and inference is performed using 900 steps. The denoising network is implemented using a 6-layer CSPNet. Optimization is done using the Adam optimizer with a learning rate of 0.001.

### C.4 Evaluation Metrics for CSP Task

We evaluate the performance of CrysLLMGen and baseline models on stable structure prediction using standard metrics proposed in prior works [10, 8], by assessing how well the generated structures match the corresponding ground truth structures in the test set. We compute the **Match Rate** and **RMSE** metrics using the StructureMatcher class from Pymatgen, which identifies the best correspondence between two structures while accounting for all material invariances. The Match Rate measures the percentage of generated structures in the test set that successfully match the ground truth structures under the thresholds: $\mathtt{stol} = 0.5$, $\mathtt{angle\_tol} = 10$, and $\mathtt{ltol} = 0.3$. The RMSE is calculated between atomic positions of the ground truth and the best-matching generated structure, normalized by $\sqrt[3]{V/N}$, where V is the lattice volume and N is the number of atoms, and averaged over all matched structures.

### C.5 Validity Metrics for Gen Task

- *Validity :* In line with previous studies [16, 8], we assess both structural and compositional validity. Structural validity represents the percentage of generated crystals with valid periodic structures, while compositional validity refers to the percentage of structures with correct atom types. A structure is considered valid if the shortest distance between any pair of atoms exceeds 0.5 Å, and its composition is deemed valid if the overall charge remains neutral, as determined by SMACT [60].

- *Coverage :* We consider two coverage metrics, COV-R (Recall) and COV-P (Precision). COV-R measures the percentage of the test set materials being correctly predicted, whereas COV-P measures the percentage of generated materials that cover at least one of the test set materials.

- *Property Statistics :* We evaluate the similarity between the generated materials and those in the test set using various property statistics, where we compute the earth mover's distance (EMD) between the distributions in element number (# Elem) and density ($\rho$, unit g/cm3).

### C.6 Stability Metrics for Gen Task

- *% Meta-Stable :* We report the percentage of structures, among 1,000 generated materials, that achieve $E^{hull} < 0.1$ eV/atom after relaxation using CHGNet.

- *% M.S.U.N :* Among the metastable structures, we report the percentage of crystal materials that are unique and novel.

- *% Stable :* Out of 1,000 generated materials, we report the percentage of structures that achieve $E^{hull} < 0.0$ eV/atom after relaxation using CHGNet. This metric reflects the thermodynamic stability of the generated materials.

- *% S.U.N :* Out of all stable structures, we report the percentage of crystal materials that are both unique and novel.

## D  Crystal Structure Prediction (CSP)

We also evaluate our proposed model on the *Crystal Structure Prediction (CSP)* task, where the goal is to generate a complete crystal structure, including atomic coordinates and lattice parameters, given only the atomic composition.

| Category | Method | Perov-5 | | Carbon-24 | | MP-20 | | MPTS-52 | |
|---|---|---|---|---|---|---|---|---|---|
| | | Match Rate ↑ | RMSE ↓ | Match Rate ↑ | RMSE ↓ | Match Rate ↑ | RMSE ↓ | Match Rate ↑ | RMSE ↓ |
| Diffusion Models | CDVAE | 45.31 | 0.1138 | 17.09 | 0.2969 | 33.90 | 0.1045 | 5.34 | 0.2106 |
| | DiffCSP | 52.02 | 0.0760 | 17.54 | 0.2759 | 51.49 | 0.0631 | 12.19 | 0.1786 |
| Flow Matching | FlowMM | 53.15 | 0.0992 | 23.47 | 0.4122 | 61.39 | 0.0566 | 17.54 | 0.1726 |
| LLMs | Llama-2-7B | 97.60 | 0.0096 | 97.21 | 0.0255 | 73.22 | 0.1546 | 39.96 | 0.1092 |
| LLM+Diffsuion | CrysLLMGen (7B) | **98.30** | **0.0066** | **98.81** | **0.0136** | **74.24** | **0.1276** | **42.16** | **0.0977** |

Table 4: Summary of results on *CSP Task* of Different Classes of Generative Models.

**Setup.** In this setting, we incorporate the atom types directly into the input prompt and use CrysLLMGen to generate the corresponding fractional coordinates and lattice structure. We conduct experiments on three benchmark datasets: Perov-5, Carbon-24, MP-20 & MPTS-52 and compare the performance against four strong baselines: CDVAE [8], DiffCSP [10], FlowMM [14], and LLaMA-2(7B) [7]. To assess the effectiveness of our model and baselines in generating stable structures, we adopt two widely used evaluation metrics, Match Rate and RMSE, as defined in prior works [8, 10], by comparing the generated structures to ground truth structures from the test set.

**Results.** We report the results in Table 4. We observe that CrysLLMGen consistently outperforms competing baseline models across all benchmark datasets, including challenging and realistic ones like MP-20. In particular, it shows significant gains over denoising-based approaches such as diffusion and flow matching models, and achieves notable improvements over standalone LLM-based methods.

## E Comparison of CrysLLMGen and other Joint Diffusion Frameworks in terms of stability

CrysLLMGen adopts a *hybrid* strategy that combines a large language model (LLM) with a symmetry-equivariant diffusion network. First, the LLM proposes a chemically plausible composition $A$; this stoichiometry is subsequently *frozen*. A second stage then performs score-based denoising of the lattice vectors $L$ and fractional coordinates $X$. In contrast, *joint* diffusion frameworks (for example, DIFFCSP) operate directly on the triplet $(A, L, X)$, requiring a single score network to denoise both discrete and continuous variables simultaneously. The latter increases the dimensionality of the sample space and introduces high-variance gradients associated with the one-hot representation of $A$.

Both pipelines optimise a likelihood-based diffusion loss, yet the LLM in CrysLLMGen is *pre-trained* on extensive corpora that embed chemical knowledge (ICSD entries, Materials Project abstracts, patents, etc.). Consequently, the LLM internalises empirical rules of charge balance, common oxidation states, and frequency-weighted formation energies that are largely *invisible* to structure-only datasets. This chemical prior endows CrysLLMGen with a selective bias toward compositions empirically known to be low-enthalpy, translating into systematically smaller energies above the convex hull, $E_{\text{hull}}$, than those produced by fully coupled diffusion.

The stability gap can be rationalised by considering the factors contributing to the energy above the convex hull for a given material structure $(A, X, L)$. $E_{\text{hull}}(A, X, L)$ can be conceptually decomposed into contributions related to the intrinsic stability of the composition $A$ itself, and the relative stability of the specific structure $(X, L)$ for that composition,

$$E_{\text{hull}}(A, X, L) = \underbrace{\Delta E_{\text{struct}}(X, L \mid A)}_{\substack{\text{Polymorph} \\ \text{energy split}}} + \underbrace{\Delta E_{\text{chem}}(A)}_{\substack{\text{Compositional} \\ \text{instability}}}. \quad (6)$$

Here, $\Delta E_{\text{struct}}(X, L \mid A)$ is the energy difference between structure $(X, L)$ and the most stable known structure for composition $A$, while $\Delta E_{\text{chem}}(A)$ represents the minimum energy above the convex hull for composition $A$. Empirical data from high-throughput density functional theory calculations across large materials databases clearly shows that $\Delta E_{\text{chem}}$ values (reflecting compositional instability) are typically an order of magnitude or more larger than $\Delta E_{\text{struct}}$ values (reflecting polymorph energy differences) [61, 62]. Because $\Delta E_{\text{chem}}$ is the dominant term, enforcing a chemically informed prior on $A$ via the LLM is the most effective route to generating thermodynamically stable structures with low $E_{\text{hull}}$. In a fully coupled model, rare yet low-$\Delta E_{\text{chem}}$ formulas are under-sampled by the standard likelihood objective, forcing the generator to explore a chemically broader and therefore statistically higher-energy compositional subspace. In other words, sampling compositions $A$ that are thermodynamically very stable ($\Delta E_{\text{chem}}(A)$) but rare in existing structural databases poses a

significant challenge, as they correspond to low-density regions in the training data. The diffusion process, driven by gradients of the log-likelihood, may struggle to explore or converge effectively in these sparse regions.

