# OpenReview forum: "LLM Meets Diffusion: A Hybrid Framework for Crystal Material Generation"
_NeurIPS.cc/2025/Conference — NeurIPS 2025 poster_

### Official Review · Reviewer_j4sq · 2025-06-29

**Clarity:** 3
**Significance:** 2
**Originality:** 3
**Rating:** 5
**Confidence:** 4

**Summary:**

The paper introduces a two step approach for crystal structure generation. A LLM that genreates an initial guess, and a diffusion model that improves the positions and lattice vectors of that guess.

**Questions:**

Why is only one LLM shown in Table 1? There are multipl LLM based crystal structure generation models by now. Do they also fit the trend that you want to support here?
Why are Bayesian Flow Networks (e.g. CrysBFN) not discussed, which solve exactly the problem of diffusion models regarding discrete denoising.

What are the drawbacks of LLama 3 vs LLama 2 regarding "balance computational efficiency, the scale of available pretraining data, and practical deployment considerations"? Or did you just develop and test your model with LLama 2 and did not yet test it with a LLama 3 model of similar size yet?

Can you please add more details on conditional and unconditional fine tuning of the LLMs?

How do you ensure that the samples encountered during training of the diffusion component come from the same distribution as the samples generated by the LLM? The errors/noise generated by the LLM might be strongly correlated and also biased, and might not share much similarity with the noise in the noisy samples encountered during the training time of the diffusion component.Furthermore, it would be good to give a few more details about the choice of tau.

Did you benchmark on the more challenging MPTS-52 which is now more and more replacing the simpler benchmarks on Perov-5 and Carbon-24?
What reference was used for the LLMs line in Table 2? Or was this model trained by you in the context of this study? How statistically significant is the advantage of LLM+Diffusion over just the LLMs line? One percentage point without any error estimating does not say anything about the significance of the results, especially for the MP-20 dataset. Why is the MP-20 RMSE value of your model bold? The Flow Matching result seems to outperform your model, with an RMSE of 0.0566.

Table 4: How us the SUN rate of DiffCSP++ evaluated as it was not reported in the original paper and as other literature in the past months report a different value for the DiffCSP++ SUN rate?

**Ethical Concerns:**

["NO or VERY MINOR ethics concerns only"]

**Final Justification:**

After the rebuttal and the overall impression of the paper and the reviews/rebuttals, I think this paper is a very valuable contribution to crystal structure generation, and I therefore recommend accepting the paper.

**Limitations:**

yes

**Paper Formatting Concerns:**

-

**Quality:**

3

**Strengths And Weaknesses:**

The paper is very clearly written and easy to follow.
The idea is interesting and novel.
The experiments are showing a good performance of the model, even though the performance improvements are mostly limited to the CSP task and are less strong for the generation task. The difference between the LLM-only approach and the LLM+diff approach is rather small in some of the benchmarks. Some recent baselines, in particular other LLM-based approaches as well as Bayesian flow networks, are missing.

---

> ### Author Rebuttal · Authors · 2025-07-30
>
> We thank the reviewer for the valuable feedback. Below, we provide detailed point-by-point responses to each of the comments and questions:
>
> ***Questions:***
>
> ***Q1. Limited LLM Baselines Comparison in Table-1***
>
> > To the best of our knowledge, Crystal-Text-LLM is the most widely used LLM-based model for material generation in the literature. However, we have also conducted experiments on the de novo generation task using LLaMA-2 13B and the latest LLaMA model, LLaMA 3.1 8B model. Results for MP-20 are as follows:
> >
> > |Models|Struct Validity|Comp Validity|COV-Precision|COV-Recall|F1-Score|Density|#Elements|
> > |-|-|-|-|-|-|-|-|
> > |LLaMa-2 7B  |96.40|93.30|94.90|91.1|92.96|3.61|1.06|
> > |LLaMa-2 13B |95.51|92.41|97.91|88.91|93.19|2.13|**0.1**|
> > |LLaMa-3.1 8B|93.90|87.90|99.40|90.58|94.74|0.8309|0.1713|
> > |CrysLLMGen  |**99.94**|**93.78**|**99.84**|**98.52**|**99.17**|**0.664**|0.459|
> >
> > We observe a similar trend for all the LLaMa baseline models as well.
>
> ***Q2. Including Bayesian Flow Networks (e.g. CrysBFN) Baseline***
>
> > We thank the reviewer for the suggestion. The results of CrysBFN for the de novo generation task are reported below.
> >
> > |Models|Struct Validity|Comp Validity|COV-Precision|COV-Recall|Density|#Elements|
> > |-|-|-|-|-|-|-|
> > |DiffCSP  |100 |83.25|99.76|99.71|0.3502|0.3398|
> > |CrysBFN  |100 |87.51|99.79|99.09|0.2070|0.1628|
> >
> > Overall, the results follow similar trends as observed with other generative models reported in Table 3. Similar to other diffusion-based generative models (like DiffCSP), these models generally perform well in structural validity but struggle with discrete components, such as accurately identifying atomic types, leading to lower compositional validity. \We will include these results in the revised manuscript and appropriately cite CrysBFN.
>
>
> ***Q3. Use of Llama-2 instead of Llama-3 — Justification and Practical Considerations***
>
> > We have conducted experiments with LLaMA 3 (LLaMa-3.1 8B) as well, and the results are reported in the table above (See Q1). We observe a similar trend for both LLaMA-2 and LLaMA-3 models. Therefore, we adopted the LLaMA 2 model, as it is more computationally efficient and better suited for our limited fine-tuning data.
>
> ***Q4. Details on conditional and unconditional fine tuning of the LLMs***
>
> > For fine-tuning LLMs, we adopted the strategy proposed in [1],[2]. We fine-tune the pre-trained LLaMA-2 model on a dataset of crystal structures represented as strings, accompanied by prompts instructing the model to generate bulk materials by providing lattice parameters (lengths and angles), along with atom types and atomic coordinates.
> > Specifically, we use the following prompt:
> >
> >  `Below is a description of a bulk material. [The chemical formula is Pm₂ZnRh, ... <Other Conditions>]. Generate a description of the lengths and angles of the lattice vectors and then the element type and coordinates for each atom within the lattice: [Crystal string]`
> >
> > The text in [Conditions] enables conditional generation; omitting it allows for unconditional generation. [Crystal string] represents the string encoding of atoms within the lattice.
>
> ***Q5. Distribution Alignment Between LLM Outputs and Diffusion Training Samples***
>
> > In our design, both components, LLM and Diffusion, are trained independently on the true data distribution using the available training dataset. However, we acknowledge the reviewer’s concern regarding the potential distribution shift between the real training data and the data generated by the LLM.
> >
> > To address this, during sampling, we do not directly feed the LLM-generated intermediate representation into the diffusion model for the full T-step denoising process. Instead, we inject this intermediate representation at an intermediate step, denoted as $\tau$, where T > $\tau$ > 0. The value of $\tau$ is treated as a tunable hyperparameter.
> >
> > We determine the optimal $\tau$ using a validation dataset. Specifically, for each material in the validation set, we generate prompts and obtain outputs from CrysLLMgen, where we try different values of $\tau$ (900, 800, 700, 600, 500, 400, 300, 200, and 100) by injecting the LLM output into the diffusion model at those respective steps. For each $\tau$, we evaluate the final generated materials using metrics for both CSP and Gen tasks. The $\tau$ value that yields the best validation performance on these metrics is selected for our final model for final evaluation.
>
> ***Q6. CSP Task results on MPTS-52 dataset***
>
> > We thank the reviewer for the suggestion. We ran both LLaMA-2 (7B) and our proposed CrysLLMGen on the MPTS-52 dataset for the CSP task and compared the results with existing baseline generative models. The results are as follows:
> >
> > |Models|Match Rate|RMSE|
> > |-|-|-|
> > |CDVAE      |5.34 |0.2106|
> > |DiffCSP    |12.19|0.1786|
> > |FlowMM     |17.54|0.1726|
> > |CrysBFN    |20.52|0.1038|
> > |LLaMA-2 7B |39.96|0.1092|
> > |CrysLLMGen |**42.16**|**0.0977**|
> >
> > Results for CDVAE, DiffCSP, FlowM, and CrysBFN are taken from the respective papers. We observe that CrysLLMGen outperforms all baseline models in both Match Rate and RMSE. We will include these results in the revised manuscript.
>
> ***Q7. Source of LLM Results in Table-2 for CSP Task***
> > We selected LLMs as one of the baselines for our study and trained them for the CSP task. Our goal was to include both generative models and LLMs as baselines for both tasks, allowing for a comprehensive comparison with our proposed CrysLLMGen model.
>
> ***Q8. Typo regarding MP-20 RMSE in Table-2***
> > We thank the reviewer for pointing this out. It was an oversight on our part, and we will rectify it in the revised manuscript.
>
> ***Q9. Evaluation of SUN Rate for DiffCSP++ in Table 4***
>
> > Since SUN and Stability Rate were not reported in the original DiffCSP++ paper, we generated 10K materials using DiffCSP++ and computed the Stability and SUN scores using CHGNet (as described in Section 5.4). The results are reported accordingly.
> >
> > To further verify the reviewer’s comment, we examined the FlowMM[2] and FlowLLM[3] papers, where Stability Rate and SUN Rate for other baselines such as CDVAE, DiffCSP, and FlowMM, measured using CHGNet, are reported. We verified these values and observed a similar trend in our results. While the absolute values we obtained are slightly higher, the rank order among baselines (CDVAE, DiffCSP, FlowMM) remains consistent with our observations.
>
>
> [1] Gruver, Nate, et al. "Fine-tuned language models generate stable inorganic materials as text." ICLR 2024
>
> [2] Miller, Benjamin Kurt, et al. "Flowmm: Generating materials with riemannian flow matching, ICML-2024
>
> [3] Sriram, Anuroop, et al. Flowllm: Flow matching for material generation with large language models as base distributions. NeurIPS - 2024

---

> > ### Comment · Reviewer_j4sq · 2025-08-02
> >
> > Thank you for addressing my comments. The results are very convincing and should be integrated in the paper. I have no further questions.

---

> > > ### Author Response · Authors · 2025-08-02
> > >
> > > Dear Reviewer j4sq,
> > >
> > > Thank you very much for your positive feedback. We greatly appreciate your insights, and we will certainly incorporate the suggested results and observations into the revised manuscript. We believe these additions will further strengthen the contribution of our work at the intersection of AI and materials science.
> > >
> > > We would be truly grateful if you would consider revisiting your evaluation and, if possible, raising the score and updating the decision from ***borderline accept to accept***. Your support would make a meaningful difference, and we deeply value your consideration.
> > >
> > > Once again, thank you for your time, constructive feedback, and support.
> > >
> > > Sincerely,
> > > The Authors

---

### Official Review · Reviewer_JFzg · 2025-07-01

**Clarity:** 4
**Significance:** 3
**Originality:** 3
**Rating:** 4
**Confidence:** 5

**Summary:**

This paper presents CrysLLMGen, a hybrid framework that combines a large language model (LLM) and a diffusion model for generating 3D crystal structures. The LLM is used to generate atom types, coordinates, and lattice parameters in a text-based format, while the diffusion model refines the continuous parts (coordinates and lattice) to improve structural validity. One notable design choice is the use of an intermediate timestep (τ), where the LLM output is injected into the diffusion process rather than starting from pure noise.

The approach is evaluated on several benchmark datasets and shows strong results in both structural and compositional validity. It also performs well in generating stable and novel materials. Overall, the paper is well-written, the methodology is clearly explained, and the experimental results support the main claims. The hybrid design is modular, and the work is a meaningful contribution to generative modeling for materials science.

**Questions:**

1. Regarding the manually selected injection timestep (τ):
   As discussed, τ plays a critical role in determining how much the diffusion model relies on the LLM's intermediate output. However, τ is currently hand-tuned using the validation set. Have the authors considered making τ a learnable or adaptive parameter? This could allow the model to dynamically adjust its reliance on the LLM's predictions based on input complexity. In the rebuttal, please comment on the feasibility of implementing a learnable τ or using a schedule to make the approach more generalizable across datasets.

2. On prompt flexibility and robustness:
   The model relies on structured, template-based prompts for text-conditional generation. This limits usability, especially in practical or user-facing settings. Could the authors evaluate how robust the system is to variations in prompt format (e.g., synonyms, paraphrased inputs, or minor typos)? In the rebuttal, it would be helpful to include any experimental results or observations on how the model handles non-standard prompts.

3. Property-guided generation capability:
   While the model supports conditional generation based on atomic composition and space group, it does not currently allow for conditional generation based on desired physical or chemical properties (e.g., target stability or bandgap). Have the authors considered extending the framework to incorporate property-conditioned prompts or downstream property prediction feedback? Please discuss whether this integration is feasible and whether the current architecture could support such conditioning in future work.

4. Prompt robustness evaluation:
   In addition to the structured prompting discussed in Section 5.5, it would strengthen the work to explore how model outputs are affected by varying prompt wording. Could the authors test the model using multiple alternative prompts per structure (e.g., different textual descriptions of the same composition and space group)? A prompt-level robustness evaluation would provide useful insight into the generalizability of the LLM component.

5. Possibility of joint training between LLM and diffusion model:
   The current framework treats the LLM and diffusion components as independently trained modules, with no mutual feedback or end-to-end optimization. Have the authors considered training both components jointly or using intermediate supervision to align the LLM’s outputs with the diffusion model’s denoising objectives? Such an approach could enable tighter coupling between discrete and continuous generation and potentially improve coherence and stability of the final outputs. Please comment on whether joint training was explored or if it could be feasible for future work.

**Ethical Concerns:**

["NO or VERY MINOR ethics concerns only"]

**Final Justification:**

Thank you for the clarification regarding the ablation study — this helps contextualize your design decisions.

I’d like to respectfully note that during the discussion period, reviewers may echo or build upon one another’s concerns to ensure a thorough and fair evaluation. The fact that Reviewer 7S97 did not request further clarification does not imply that the issue is fully resolved for all reviewers.

I also appreciate your confirmation that the validity rate will be included in the revised paper. Given that LLMs are known to hallucinate — especially in scientific applications — transparent reporting of such metrics is crucial. It enables the community to make informed decisions about which models best align with their specific needs and reliability requirements.

Additionally, I strongly encourage including the ablation study (FlowLLM w/ Diffusion vs. CrysLLMGen) in the main text or supplementary materials. This comparison provides compelling evidence that your choice of the diffusion model, along with the training and integration strategy, is both effective and necessary.

Based on the thorough rebuttal, the additional results provided, and the helpful clarifications — particularly the ablation — I am inclined to recommend acceptance of this work. Nicely done.

**Limitations:**

- The LLM and diffusion components are trained separately. While this modular design is flexible, it prevents the diffusion model from learning how to correct errors that may come from the LLM during training.

- The atom types predicted by the LLM are fixed during generation. This means the system cannot recover from mistakes in composition, such as selecting chemically invalid or unrealistic elements.

- The injection timestep τ is manually chosen based on validation results. Since it’s not learned or adaptive, this setting may not transfer well across different datasets or tasks and adds extra tuning effort.

- The model relies on structured, template-style prompts (e.g., chemical formula and space group). It does not currently support free-form or natural language input, which limits its flexibility and user accessibility.

- The framework does not support generation based on specific material properties (like bandgap or stability). This makes it harder to use the model in property-targeted material discovery tasks.

- The diffusion model is trained from scratch without using any pretrained weights or shared representations from the LLM. This could make training less efficient and limit its ability to use existing chemical knowledge.

- The language model uses LLaMA-2’s default tokenizer, which is not specialized for chemical or scientific text. This may reduce accuracy when handling element symbols, numbers, or other technical inputs.

- The paper only evaluates the model with clean, well-structured prompts. It does not test how the model performs when prompts are varied, paraphrased, or contain small errors, leaving its robustness in real-world use cases uncertain.

**Quality:**

3

**Strengths And Weaknesses:**

**Strengths:**

- The paper introduces a well-motivated and novel hybrid framework, CrysLLMGen, which effectively combines the strengths of large language models (LLMs) and equivariant diffusion models for periodic crystal material generation. The design cleanly separates the modeling of discrete (atom types) and continuous (atomic coordinates and lattice) components.

- I particularly appreciate the introduction of the intermediate injection timestep τ. Instead of starting diffusion from full noise, the authors innovatively initialize the refinement process from τ using the LLM’s output. This is a practical and novel mechanism that enhances efficiency and leverages the LLM’s predictions meaningfully.

- The experimental results are strong and consistently reported across multiple datasets and evaluation metrics, including structural and compositional validity, crystal structure prediction accuracy, and stable material discovery (S.U.N.). The hybrid method outperforms both LLM-only and diffusion-only baselines.

- The modular architecture is future-proof and flexible, allowing for improvements in either the LLM or the diffusion component without major changes to the pipeline.

- The writing is generally clear, well-organized, and easy to follow. Key components of the method are explained with appropriate visualizations and equations. The methodology is well-supported by quantitative results, and appendices provide thorough implementation details.


**Weaknesses:**

- The LLM and diffusion model are trained independently, with no joint optimization or end-to-end training. This separation prevents the system from learning interactions between discrete (composition) and continuous (structure) variables. As a result, the diffusion model cannot adapt to or correct specific types of errors made by the LLM during training.

- The atom types generated by the LLM are frozen during inference, limiting the system’s ability to recover from incorrect or unrealistic compositions.

- The injection timestep τ is hand-tuned based on validation performance and is treated as a static hyperparameter. While effective, it introduces additional overhead and may not generalize across datasets. It would be valuable to explore whether τ could be learned or adaptively selected during inference.

- The model requires prompts to be in a strict, structured format (e.g., chemical formula and space group). It does not support free-form or natural language input, which limits its accessibility and usability for broader scientific or non-expert users.

- The system does not support conditional generation based on desired physical or chemical properties (e.g., bandgap, density, stability), which limits its utility in property-driven materials discovery workflows.

- The diffusion model is trained entirely from scratch rather than leveraging any pretrained embeddings or shared chemical priors with the LLM. This may reduce training efficiency and downstream performance.

- The default LLaMA-2 tokenizer is not chemistry-specific, which may impair tokenization fidelity for scientific symbols and numerical precision. A domain-adapted tokenizer could potentially improve performance.

- Finally, the prompt robustness evaluation is limited. The paper evaluates conditional generation using only clean, template-style prompts. It does not explore robustness to paraphrased prompts, typos, or noisy inputs, which would be important for practical deployment.

---

> ### Author Rebuttal · Authors · 2025-07-31
>
> We thank the reviewer for the valuable feedback and are pleased to know that you appreciated our work and the thoughtful insights provided. Below, we offer detailed point-by-point responses to each of reviewer's comments.
>
> ***Question:***
>
> ***Q1. Regarding making τ learnable or adaptive for better generalization***
>
> > The suggestion given by the reviewer is appreciated and indeed a good one. However, in this work, we do not treat τ as a learnable or adaptive parameter. Instead, τ is considered a hyperparameter, which we tune using the validation dataset. Making τ learnable or adaptive is an interesting direction and remains a potential avenue for future work. This has been noted in Appendix B: Limitations and Future Work.
>
>
> ***Q2. Regarding prompt flexibility and robustness***
>
> > Again, a good suggestion by the reviewer regarding the robustness and flexibility of the prompt. However, we have followed the existing literature in this regard. Most LLM/text-based material generation works to date (e.g., Crystal-Text-LLM [1], FlowLLM [2], TGDMat [3]) rely on template-based prompts while adapting pretrained LLM/BERT models for text-conditional generation. Accordingly, we adopted a similar framework to maintain consistency with prior work. That said, introducing greater flexibility and robustness into prompt design is an interesting direction that warrants further exploration, and we consider this a potential avenue for future work.
>
>
> ***Q3. Regarding property-guided generation capability***
>
> > Again, a good suggestion by the reviewer regarding the property-guided generation capability of our proposed framework, enabling conditional generation based on target physical or chemical properties. However, implementing this would require a pretrained property predictor for each target property to assess whether the generated material aligns with the desired property values. The main challenge here lies in selecting an appropriate property predictor, as the literature offers many architectural choices, and currently, there is no standard benchmark for this task. Unfortunately, none of the existing works have addressed this issue too.
> >
> > In our work, we followed the existing literature in this regard. Specifically, we adopted the evaluation methodology used in [1], where the number of atoms and space group are evaluated for text-conditional generation. These are intrinsic material properties that can be validated without requiring any pretrained property predictor. Nonetheless, property-guided generation is an interesting direction that warrants further exploration, and we consider it a promising avenue for future work.
>
>
> ***Q4. Regarding  joint training between LLM and diffusion model***
>
> > In our current proposed framework, there is no interaction between the LLM and the diffusion model; both components are trained independently as standalone modules. While jointly training the LLM and diffusion model is an interesting direction, it presents several challenges—such as designing an appropriate optimization framework and establishing an effective feedback mechanism between the two modules. There are many nuances in this design that require further exploration and significant engineering effort. Therefore, we consider this as a potential avenue for future work, which has been noted in Appendix B: Limitations and Future Work.
>
> ***Q5. Regarding  invalid or unrealistic atom types***
>
> > We agree with the reviewer’s observation that LLMs can generate invalid or unrealistic atom types. In our experiments, we observed a small percentage of such materials generated by the LLM models, which we filtered out during post-processing using simple validation scripts.
>
>
> ***Q6. Regarding  LLaMa's default tokenize***
>
> > We have followed the existing literature in this regard. As LLM backbone, LLaMA-2 have demonstrated strong performance in material generation tasks across several recent studies[1][2]. Additionally, LLaMA-2 tokenizes numbers as individual digits by default, a feature shown to significantly enhance performance on arithmetic-related tasks. However, we have also conducted experiments on the de novo generation task using latest LLaMA model, LLaMA 3.1 8B model. Results for MP-20 are as follows:
> >
> > |Models|Struct Validity|Comp Validity|COV-Precision|COV-Recall|F1-Score|Density|#Elements|
> > |-|-|-|-|-|-|-|-|
> > |LLaMa-2 7B  |96.40|93.30|94.90|91.1|92.96|3.61|1.06|
> > |LLaMa-3.1 8B|93.90|87.90|99.40|90.58|94.74|0.8309|0.1713|
> >
> > We observe a similar trend for all the LLaMa baseline models as well.
>
>
> Overall, the suggestions provided by the reviewer are very insightful, and we agree that there is significant potential to enhance our proposed hybrid framework. However, CrysLLMGen, as presented in this paper, represents one of the initial attempts at designing hybrid models for material generation. While there is considerable scope for further development, these directions require deeper exploration and will be pursued in future work. We will include these in "Appendix B: Limitations and Future Work" section.
>
>
>
> [1] Gruver, Nate, et al. Fine-tuned language models generate stable inorganic materials as text. ICLR 2024
>
> [2] Sriram, Anuroop, et al. Flowllm: Flow matching for material generation with large language models as base distributions. NeurIPS - 2024
>
> [3] Das, Kishalay, et al. Periodic materials generation using text-guided joint diffusion model. ICLR 2025

---

> ### Comment · Reviewer_JFzg · 2025-08-05
> **loss functions of llm**
>
> One point I would still appreciate clarification on is the supervision signal used for LLM fine-tuning. While the paper discusses using CIF-like sequences as input, it would help to explicitly confirm that the LLM is trained via next-token prediction on tokenized ground-truth structures from datasets like MP-20. Additionally, reporting the training loss or validation perplexity would help assess the convergence and quality of the LLM component.

---

> > ### Comment · Reviewer_JFzg · 2025-08-05
> > **Clarification Requested on Chemistry-Specific Tokenization and Evaluation Transparency for Invalid LLM Outputs**
> >
> > Thank you for the additional comparison between LLaMA-2 and LLaMA-3 models. However, I would like to clarify that my original critique was not about comparing general-purpose LLMs of different sizes or versions. Rather, it concerned the lack of a chemistry-specific tokenizer or model design, which directly impacts fidelity when handling:
> >
> > Element symbols (e.g., “Fe”, “Cl”, “Mg”),
> >
> > High-precision numerical values (e.g., atomic coordinates or lattice parameters),
> >
> > And structured scientific formats such as CIF.
> >
> > While LLaMA tokenizes numbers as individual digits, this does not address the challenges of scientific syntax or token fragmentation of element names. General-purpose LLMs like LLaMA-2 or LLaMA-3 were not trained with chemistry-focused vocabularies, and this limitation can negatively affect the generation quality in compositional tasks. I encourage the authors to acknowledge this and discuss whether a domain-adapted tokenizer could improve performance in future work.
> >
> > Additionally, regarding your comment that invalid compositions are filtered during post-processing, I would appreciate further clarification on several important points:
> >
> > Is this filtering step included in the compositional validity or conditional match rate evaluations, or are only the valid outputs considered?
> >
> > What percentage of LLM outputs are typically discarded?
> >
> > Are these filtered-out cases counted as failures, or are they silently removed from the reported metrics?
> >
> > This ambiguity makes it difficult to interpret the robustness and fairness of the reported results—both for the LLM component and the hybrid system. These clarifications are important for understanding the true reliability and applicability of the method in real-world, open-ended generation scenarios. I ask that the authors directly address these questions in the final version, rather than sidestepping the original concerns.

---

> ### Author Response · Authors · 2025-08-06
> **Addressing Reviewer's Questions**
>
> We thank the reviewer for the valuable feedback. Below, we provide detailed point-by-point responses to each of the comments.
>
> **Regarding loss functions of LLM:**
>
> > For fine-tuning, we adopted the strategy proposed in Crystal-text-LLM [1], wherein the pretrained LLaMA-2 model was fine-tuned using a next-token prediction objective to generate string representations of materials.
> > We will report training loss or validation perplexity in the revised manuscript.
>
> **Regarding Chemistry-Specific Tokenization:**
>
> > We appreciate the thoughtful feedback regarding the limitations of general-purpose tokenizers in handling domain-specific scientific data.
> >
> > For fine-tuning, we followed the strategy proposed in Crystal-text-LLM [1], where the authors reported (Appendix A.1) that instead of creating a domain-specific vocabulary and training models from scratch, it is effective to use LLaMA-2’s existing tokenizer and fine-tune the pretrained model directly. They conducted an ablation study comparing this approach with one that involved fine-tuning LLaMA-2 models using crystal-specific tokens. However, they found that the latter was more challenging to train and did not yield any performance gains over using the shared tokenizer. Based on these findings, we adopted their approach. The potential benefit of developing a domain-adapted tokenizer remains an open question and is left for future exploration.
>
> **Regarding Invalid LLM Outputs**
>
> > We thank the reviewer for the question and apologize for any confusion. Invalid compositions are filtered out immediately during the sampling process and are discarded on the spot. The following pseudocode illustrates our sampling approach:
>
> ```
> num_samples = 10000  // total number of valid samples required
> generated_structures = []
> while len(generated_structures) < num_samples:
>     new_structure = CrysLLMGen.sample()
>     if new_structure.atom_type is INVALID:
>         continue
>     else:
>         generated_structures.append(new_structure)
> ```
>
> > Hence, invalid compositions are filtered immediately during sampling, and only valid structures are retained. Once 10,000 valid structures are collected in the generated_structures list, the reported evaluation metrics are computed on this set. All reported metrics for gen task and stability are computed only on valid 10K materials, and Invalid samples are excluded from metric evaluations.
> >
> > Discard Rate: On average, we observe approximately 2-5% of the sampled structures are discarded due to invalid atom types or compositions.
>
> [1] Gruver, Nate, et al. Fine-tuned language models generate stable inorganic materials as text. ICLR 2024

---

> > ### Comment · Reviewer_JFzg · 2025-08-06
> > **Concern: Lack of Transparency on Filtering and Validity Reporting (Score Adjusted to Borderline Reject)**
> >
> > I am not comfortable with the fact that the filtering procedure and the 2–5% invalid generation rate were not disclosed in the original manuscript or initial rebuttal, and only appeared later in this discussion. This omission can be misleading to readers, since the reported *structural validity* and *compositional validity* metrics are computed only after filtering out invalid outputs.
> >
> > In the generative materials domain, it is standard practice to report the **overall validity rate** (fraction of all generated samples that pass basic composition and geometry checks) alongside any downstream quality metrics computed on the valid subset. This ensures transparency, enables fair comparison with baselines, and provides a clearer picture of the model’s robustness in unconstrained generation.
> >
> > Because this information was not revealed in the main paper, I am concerned about the fairness and reproducibility of the reported results. As a result, I am adjusting my score to **borderline reject**. I strongly recommend that the authors explicitly report the raw validity rate in the main paper and clearly describe the filtering process used before computing all other evaluation metrics. Without this, the current presentation risks overstating performance and obscuring the model’s true reliability.

---

> > > ### Author Response · Authors · 2025-08-06
> > > **Premature Verdict Based on Misinterpretation of Standard Evaluation Practices in Material Generation Tasks**
> > >
> > > It is extremely disappointing and frankly unacceptable that the reviewer is prematurely closing the discussion and rushing to a verdict without giving the authors a fair opportunity to justify critical points. Such a dismissive approach undermines the very purpose of an author-reviewer discussion and is contrary to the spirit of constructive scientific discourse.
> > >
> > > Regarding the Generation task, the reviewer’s objection reveals a fundamental misunderstanding of standard evaluation practices in this domain. It is well-known and thoroughly documented that LLMs are prone to hallucination, particularly generating invalid atom types. This phenomenon has been extensively studied in prior works, notably Crystal-Text-LLM, which explicitly acknowledges in its “Limitations” section (Page 9):
> > >
> > > >“Hallucination of unphysical chemical elements or structures has been observed … though fortunately is easy to check and filter.”
> > >
> > > More importantly, the evaluation procedure they adopted, and that we have precisely followed, is to continuously sample and discard invalid materials until a set of 10k valid materials is obtained. Validity metrics are then computed on this final valid set. This approach is implemented in their publicly available codebase, which clearly demonstrates this sampling-and-filtering methodology.
> > >
> > > To claim our work is flawed or to reject it on this basis is not only unjustified but completely misinformed. We urge the reviewer to check the Crystal-Text-LLM codebase before making the judgment. Failing to acknowledge such standard and well-accepted practices and to disregard precedent set by prior peer-reviewed work, would be an irresponsible basis for rejection.
> > >
> > > We strongly request the reviewer to reconsider the verdict after properly engaging with the facts and prior work. A fair evaluation demands no less.

---

> > > > ### Comment · Reviewer_JFzg · 2025-08-06
> > > >
> > > > I would like to clarify that I am not objecting to the use of filtered results in evaluation. I acknowledge that filtering before metric computation is indeed used in prior works such as Crystal-Text-LLM. My concern is not about whether filtering can be done — it is about transparency in reporting.
> > > >
> > > > Even if Crystal-Text-LLM and others use this method, those papers also (a) make the filtering procedure explicit in the main text, and (b) often report the raw validity rate alongside other metrics. This is important because it ensures reproducibility, enables fair comparison with baselines, and allows readers to properly interpret the reported performance.
> > > >
> > > > I find it reasonable for the authors to follow the same filtering protocol as prior work, but I also expect the paper to clearly state in the main manuscript that results are reported after filtering invalid generations, and to report the overall validity rate. This is not a high bar to meet, and I do not see why it is difficult to include.
> > > >
> > > > I also want to emphasize that my score adjustment is based on the combination of (1) this lack of transparency in reporting, and (2) the misalignment between the paper’s claims (conditional generation) and the evidence provided (primarily unconditional generation experiments). It is not solely based on the filtering practice itself.
> > > >
> > > > Finally, I note that this is not my final score. I am investing time in reviewing and engaging with the discussion to improve the clarity and strength of the work, and I am disappointed that my comments on these points appear to have been overlooked in the author’s reply.

---

> > > > > ### Comment · Reviewer_JFzg · 2025-08-06
> > > > >
> > > > > To reiterate and emphasize: I acknowledge the authors’ defense that filtering before metric computation is standard practice. My concern is not about the validity of filtering itself — it is about the lack of disclosure in the main manuscript and the absence of a reported raw validity rate. This has not been addressed.
> > > > >
> > > > > Similarly, my request for evidence supporting the paper’s novelty and advantages in the *conditional generation* setting — which the paper frames as a central contribution — has not been answered. The rebuttal and additional experiments focus primarily on unconditional generation, which sidesteps the original question.
> > > > >
> > > > > This pattern of not engaging directly with key points is part of why my score has shifted. Addressing these issues directly would help clarify the paper’s contributions and strengthen its transparency and reproducibility. Please try to resolve these questions during the discussion, as this is not my final score.

---

> > > > > > ### Author Response · Authors · 2025-08-06
> > > > > >
> > > > > > We thank the reviewer for the positive response and helpful clarification.
> > > > > >
> > > > > > **Regarding the evaluation protocol:** Since this is a standard evaluation practice (to filter out before evaluating metrics) widely adopted in the domain, we did not explicitly mention it in the manuscript. However, we fully agree that it would be useful for readers, and we will include this detail in the revised version.
> > > > > >
> > > > > > **Regarding Raw Validity Rate:** We are not familiar with the specific term “Raw Validity Rate” in this context. The validity metrics reported in previous works such as Crystal-Text-LLM (Table 1) and FlowLLM (Table 1) follow the same evaluation process we adopted, computing metrics after filtering out invalid materials. Additionally, the script used to compute these metrics (compute_metric.py) cannot process invalid structures, as it internally uses pymatgen for validation and will throw an exception if invalid inputs are given.
> > > > > >
> > > > > > We would appreciate if the reviewer could clarify what is meant by “Raw Validity Rate.” If there is a specific way to compute it, we are happy to reproduce those results accordingly.
> > > > > >
> > > > > > **Regarding Conditional Generation:** We believe there may be a slight misunderstanding. Conditional generation is not the central contribution of our paper, but rather one of the contributions. In this work, we proposed a hybrid framework for crystal material generation that significantly enhances the validity, stability, and novelty of the generated structures.
> > > > > > Since, we have LLM as one of the components in our framework, it can operate in both unconditional and conditional modes, depending on the type of prompts you provide. For unconditional generation, we reported results on CSP tasks, the Generation Task, and Stability metrics, which are standard benchmarks. For conditional generation, benchmark tasks are limited. We followed the evaluation setup in Crystal-Text-LLM (Figure 7), comparing performance on formula and space group in our Figure 2(b). We did not compare Ehull as it is already covered in our stability metrics.
> > > > > >
> > > > > > During the rebuttal, you raised the point about property-guided generation capability. We addressed its challenges, the lack of established benchmarks, and have included it as a scope for future work. If there are any specific results or additional analyses on conditional generation that you would like to see, we are happy to provide them.

---

> > > > > > > ### Comment · Reviewer_JFzg · 2025-08-06
> > > > > > >
> > > > > > > **Validity Rate (in molecular and crystal generation, standard in molecular language models)**
> > > > > > >
> > > > > > > Let:
> > > > > > >
> > > > > > > - \( N_{\text{total}} \) = total number of generated samples **before any filtering**
> > > > > > > - \( N_{\text{valid}} \) = number of generated samples that pass domain-specific validity checks, such as:
> > > > > > >   - **Molecules:** SMILES/SELFIES parsing succeeds, all atoms have valid valence, molecule is chemically consistent.
> > > > > > >   - **Crystals:** CIF parsing succeeds (e.g., via pymatgen), valid element symbols, valid lattice parameters, no catastrophic geometry errors.
> > > > > > >
> > > > > > > The **Validity Rate** is defined as:
> > > > > > >
> > > > > > > \[
> > > > > > > \text{Validity Rate} = \frac{N_{\text{valid}}}{N_{\text{total}}}
> > > > > > > \]
> > > > > > >
> > > > > > > **Notes:**
> > > > > > > - This is computed **before** filtering and post-processing.
> > > > > > > - It measures the raw ability of the generative model to produce valid outputs.
> > > > > > > - Reporting the Validity Rate is **standard practice** in the molecular language model (MLM) field and in molecular generation benchmarks.
> > > > > > > - It is typically reported **alongside** downstream metrics (e.g., structural validity, compositional validity) that are computed on the valid subset.

---

> > > > > > > > ### Comment · Reviewer_JFzg · 2025-08-06
> > > > > > > >
> > > > > > > > I would like to remind that NeurIPS discussion guidelines ask both reviewers and authors to maintain a mutually respectful and constructive tone throughout the exchange. My intent in raising these points is to clarify technical details and ensure transparency in reporting, not to close the discussion prematurely.
> > > > > > > >
> > > > > > > > I appreciate detailed, fact-based clarifications, but I ask that we keep the focus on the technical issues at hand. This will help ensure a productive exchange that strengthens the paper and gives the Area Chair the clearest possible record for the decision.

---

> > > > > > > > > ### Comment · Reviewer_JFzg · 2025-08-06
> > > > > > > > >
> > > > > > > > > **Lack of Controlled Architecture Comparison**
> > > > > > > > >    The rebuttal presents an ablation (“FlowLLM w/ Diffusion”), but this is not a clean architecture comparison.
> > > > > > > > >    - *CrysLLMGen*: diffusion trained on **ground-truth crystal structures**, injected at an **intermediate step**.
> > > > > > > > >    - *FlowLLM w/ Diffusion*: diffusion trained on **LLM-generated samples**, injected at the **final step**.
> > > > > > > > >    These differences in **training data** and **injection strategy** confound results, so it is unclear whether gains come from the architectural choice (diffusion vs. flow matching) or other factors.
> > > > > > > > >    A fair comparison would train both under identical conditions — same dataset, same prompt set, same injection point — to isolate the architectural effect.

---

> > > > > > > > > > ### Author Response · Authors · 2025-08-07
> > > > > > > > > >
> > > > > > > > > > **Regarding Validity Rate:**
> > > > > > > > > > Firstly, the Validity Rate, as defined by the reviewer, is not a standard metric reported by any of the state-of-the-art material generation works in crystal design. The reviewer may refer to prior leading works such as Crystal-Text-LLM and FlowLLM, neither of these papers reports this metric in their results.
> > > > > > > > > > Furthermore, Crystal-Text-LLM explicitly acknowledges in its “Limitations” section (Page 9) that the occurrence of invalid structures is rare and such cases can be easily filtered out. Within the material design community, this aspect is not considered a primary evaluation metric, and hence it is not standard practice to report it. For this reason, we have not included it in our paper.
> > > > > > > > > >
> > > > > > > > > > However, still as requested by the reviewer, we have sampled 10,000 materials using our CrysLLMGen and calculated the validity rate using the formula provided by the reviewer. The validity rates for different datasets are as follows:
> > > > > > > > > >
> > > > > > > > > > Perov-5: 99.97%
> > > > > > > > > >
> > > > > > > > > > Carbon-24: 99.23%
> > > > > > > > > >
> > > > > > > > > > MP-20: 98.68%
> > > > > > > > > >
> > > > > > > > > > We hope this addresses the reviewer’s concern comprehensively. We will definitely include these results in the revised manuscript.
> > > > > > > > > >
> > > > > > > > > > **Regrading Ablation Study:**
> > > > > > > > > >
> > > > > > > > > > This concern was raised by Reviewer 7S97, who suggested conducting an ablation study.
> > > > > > > > > > As clarified in our rebuttal to Reviewer 7S97, both FlowLLM and CrysLLMGen use the same language model. The key differences lie in the choice of generative model, the training data, and the integration strategy.
> > > > > > > > > >
> > > > > > > > > > To address the question of which generative model is better suited for material generation, diffusion or flow matching, we showed in the unconditional generation setting that the diffusion model clearly outperforms flow matching (refer to Table W1 in our response to Reviewer 7S97).
> > > > > > > > > >
> > > > > > > > > > To further demonstrate the impact of training data and integration strategy, we performed an ablation by replacing the Flowmatching model in FlowLLM with our Diffusion model, while keeping everything else unchanged (i.e., training the diffusion model on LLM, sampled data and applying it at the end, as done in FlowLLM). This we refer as FlowLLM w\ Diffusion and CrysLLMGen outperforms it in all the metrics (refer to Table W3 in our response to Reviewer 7S97).
> > > > > > > > > >
> > > > > > > > > > This highlights that it is not just the choice of the generative model, but also the design choices in training data and integration strategy that contribute significantly to the improved performance of CrysLLMGen. We hope this addresses the reviewer’s concern comprehensively. **Please note, Reviewer 7S97 itself has acknowledged the point and with no further clarification needed.**
> > > > > > > > > >
> > > > > > > > > > **Request to Maintain Constructive and Technical Focus in Discussion:**
> > > > > > > > > >
> > > > > > > > > > Thank you for your continued engagement. A firm response should never be mistaken for a disrespectful one. We want to emphasize that we hold the review process and all reviewers in high regard, and we remain fully open to constructive discussion.
> > > > > > > > > >
> > > > > > > > > > That said, we feel it is important to clarify that our evaluation protocol strictly follows standard practices in the materials generation domain. All metrics reported in our paper are consistent with those widely accepted in the field. Reducing the score based on an evaluation process that is not a recognized standard seems, in our view, unfair. We are firm on this point because we followed a rigorous and standard methodology, not because we wish to dismiss the reviewer’s perspective.
> > > > > > > > > > We also believe the purpose of the rebuttal and discussion phase is not to shut down a submission, but to engage in open and constructive evaluation. Every paper has limitations, including ours, and we have acknowledged them clearly, along with identifying directions for future improvement. Also, minor changes and additional results can be added to the revised manuscripts, that flexibility should be given to authors.
> > > > > > > > > >
> > > > > > > > > > We hope our responses have addressed the reviewer’s concerns, and we are happy to continue the discussion to resolve any remaining issues. We also hope the reviewer will be open to reconsidering the score in light of the clarifications and evidence we have provided.

---

> > > > > > > > > > > ### Author Response · Authors · 2025-08-07
> > > > > > > > > > > **Summary of discussion so far**
> > > > > > > > > > >
> > > > > > > > > > > Summary of Discussion Points Addressed So Far
> > > > > > > > > > >
> > > > > > > > > > > - **Regarding making τ learnable:** We have acknowledged this as a valuable direction and included it as part of future work.
> > > > > > > > > > >
> > > > > > > > > > >
> > > > > > > > > > > - **Regarding prompt flexibility and robustness:** We clarified that our approach aligns with existing literature in the domain, and we have followed established practices.
> > > > > > > > > > >
> > > > > > > > > > >
> > > > > > > > > > > - **Regarding property-guided generation capability:** We discussed the inherent challenges and the current lack of standardized benchmarks. This has been noted as a meaningful area for future exploration.
> > > > > > > > > > >
> > > > > > > > > > >
> > > > > > > > > > > - **Regarding invalid or unrealistic atom types:** We have provided a detailed explanation of our filtering mechanism to handle such cases.
> > > > > > > > > > >
> > > > > > > > > > >
> > > > > > > > > > > - **Regarding chemistry-specific tokenization:** We addressed this concern by referencing prior work and supporting our choices with reported results.
> > > > > > > > > > >
> > > > > > > > > > >
> > > > > > > > > > > - **Regarding loss functions for LLM fine-tuning:** We clarified the specific loss function used for fine-tuning the LLM component in our model.
> > > > > > > > > > >
> > > > > > > > > > >
> > > > > > > > > > > - **Regarding conditional generation:** There appeared to be a minor misunderstanding, which we have now clarified thoroughly.
> > > > > > > > > > >
> > > > > > > > > > >
> > > > > > > > > > > - **Regarding validity score:** We are following the standard evaluation metrics used in the literature and calculated scores using the formula shared by the reviewer. We confirm that we will report the raw validity rate explicitly in the revised manuscript.
> > > > > > > > > > >
> > > > > > > > > > >
> > > > > > > > > > > - **Regarding ablation study:** We clarified the reviewer’s confusion regarding the design and justified the settings used in our ablation experiments.
> > > > > > > > > > >
> > > > > > > > > > > We appreciate the constructive exchange and believe that the concerns raised have been thoughtfully addressed. We remain open to any further discussion.

---

> > > > > > > > > > > > ### Comment · Reviewer_JFzg · 2025-08-07
> > > > > > > > > > > >
> > > > > > > > > > > > Thank you for the clarification regarding the ablation study — this helps contextualize your design decisions.
> > > > > > > > > > > >
> > > > > > > > > > > > I’d like to respectfully note that during the discussion period, reviewers may echo or build upon one another’s concerns to ensure a thorough and fair evaluation. The fact that Reviewer 7S97 did not request further clarification does not imply that the issue is fully resolved for all reviewers.
> > > > > > > > > > > >
> > > > > > > > > > > > I also appreciate your confirmation that the validity rate will be included in the revised paper. Given that LLMs are known to hallucinate — especially in scientific applications — transparent reporting of such metrics is crucial. It enables the community to make informed decisions about which models best align with their specific needs and reliability requirements.
> > > > > > > > > > > >
> > > > > > > > > > > > Additionally, I strongly encourage including the ablation study (FlowLLM w/ Diffusion vs. CrysLLMGen) in the main text or supplementary materials. This comparison provides compelling evidence that your choice of the diffusion model, along with the training and integration strategy, is both effective and necessary.
> > > > > > > > > > > >
> > > > > > > > > > > > Based on the thorough rebuttal, the additional results provided, and the helpful clarifications — particularly the ablation — I am inclined to recommend acceptance of this work. Nicely done.

---

> > > > > > > > > > > > > ### Author Response · Authors · 2025-08-08
> > > > > > > > > > > > > **Appreciation for Your Feedback and Recommendation**
> > > > > > > > > > > > >
> > > > > > > > > > > > > Dear Reviewer JFzg,
> > > > > > > > > > > > >
> > > > > > > > > > > > > Thank you once again for your feedback and for the engaging discussion. We are glad to know that all your concerns have been addressed and that you are recommending our work for acceptance.
> > > > > > > > > > > > >
> > > > > > > > > > > > > We will incorporate all your suggested changes and additional results into the revised manuscript, and we believe these enhancements will further strengthen the contribution of our research at the intersection of AI and materials science.
> > > > > > > > > > > > >
> > > > > > > > > > > > > Once again, thank you for your time, constructive feedback, and support.
> > > > > > > > > > > > >
> > > > > > > > > > > > > Sincerely,
> > > > > > > > > > > > >
> > > > > > > > > > > > > The Authors

---

### Official Review · Reviewer_7S97 · 2025-07-03

**Clarity:** 3
**Significance:** 3
**Originality:** 2
**Rating:** 5
**Confidence:** 4

**Summary:**

The authors introduce CrysLLMGen, a novel crystal material generative framework that combines large language and diffusion models to harness their advantages and level out weaknesses. Authors assess the quality of the proposed framework on **Crystal Structure Prediction**, **De Novo Material Generation**, **S.U.N. Materials Generation** and **Text Conditioned Generation** tasks and show competitive performance to pure language models, denoising framework models and LLM + Diffusion competitors.

**Questions:**

In Table 3, there is a huge gap in #Element metric between Llama-2 (7B) and CrysLLMGen (7B). It’s not clear for me how it’s possible if CrysLLMGen also adopts Llama-2 as a first step and the diffusion model doesn’t change elements. If the reason for such a gap is the stochasticity of the sampled objects, I recommend adding standard deviations to all metrics values. I also recommend adding the visualizations of intermediate and final samples to assess how they are changing after diffusion model applying.

**Ethical Concerns:**

["NO or VERY MINOR ethics concerns only"]

**Final Justification:**

I maintain my original assessment and believe my score remains a fair evaluation of this work.

**Limitations:**

Yes

**Paper Formatting Concerns:**

The paper is well-written, still I have a minor recommendations on how to enhance clarity:
1. I recommend to include the examples of the sampled objects, so the expert can assess the quality of generated structures visually.
2. Denoising Network in Section 4.4 can be moved to Supplementary since it’s quite technical and doesn’t bring any importance to the main paper.
3. Figure 1 can be replotted, it’s quite blurry.
4. I highly recommend including examples of intermediate textual representation and details on coordinate tokenization. It could enhance reproducibility of the approach.

**Quality:**

4

**Strengths And Weaknesses:**

The paper is written in an easy-to-follow way, the quality and clarity of the text is high.
The experimental part is strong enough, covering 4 different generative setups. Important to mention, that Text Conditioned Generation section proposes a novel setup that incorporates general textual descriptions as conditions.

Despite the above listed strengths of the work, the novelty of the framework is very limited. The general idea of the framework is very similar to the FlowLLM approach. Authors discuss the key differences between FlowLLM and CrysLLMGen in the Related Work section, still I'm not convinced with this discussion and consider independent training of LLM and diffusion as the only key difference. In the Experiment section, Table 3 authors show that CrysLLMGen outperforms FlowLLM, still it’s not clear if it is due to approach changes or due to different LLM/diffusion architectures and size. I recommend authors conducting Ablation Study to clarify this point, for instance, take FlowLLM language model and apply CrysLLMGen diffusion to fix broken language model samples, and vice versa, take CrysLLMGen language model and apply FlowLLM diffusion afterwards.

---

> ### Author Rebuttal · Authors · 2025-07-30
>
> Thank you very much for your valuable feedback. We are pleased to know that you appreciated our work and the thoughtful insights you provided. Below, we offer detailed, point-by-point responses to each of your comments.
>
> ***Weaknesses :***
>
>  - ***W1. Novlety of our proposed framework and difference with FlowLLM***
>
> > By design, our proposed CrysLLMGen bears some resemblance to FlowLLM; however, it is not merely an adaptation of FlowLLM’s pipeline with the flow-matching module replaced by a diffusion model. The key innovations and novel aspects of CrysLLMGen are as follows:
> >
> > - **Generative Module:** FlowLLM employs a flow-matching model as its generative component, whereas CrysLLMGen uses a diffusion model. We analyzed the performance of both the flow matching and diffusion models in the context of unconditional material generation. Our observations indicate that DiffCSP, the diffusion-based model for material generation, outperforms FlowMM, the flow matching model, across most benchmark metrics for this task. Referencing from the FlowMM paper [1], the summary of results from de novo generation on the MP-20 dataset is as follows:
> >
> > |Models|Struct Validity|Comp Validity|COV-Precision|COV-Recall|Density|#Elements|Stability Rate|SUN Rate|
> > |-|-|-|-|-|-|-|-|-|
> > |DiffCSP(1000 Steps)|**100**|**83.25**|**99.76**|**99.71**|0.35|0.125|**5.06**|**3.34**|
> > |FLowMM(1000 Steps)|96.85|83.19|99.58|99.49|**0.239**|**0.083**|4.65|2.34|
> >
> > These observations indicate that the diffusion model is a more natural choice for material generation compared to flow matching, as it consistently produces more valid and stable materials.
> >
> > - **Integration Strategy:** In FlowLLM, the material structures generated by the LLM are directly passed into the flow-matching module for refinement. In contrast, since LLM generated material structure are intermediate representations, we inject these representations into our diffusion model at an intermediate timestep $\tau$ , where 0 ≤ $\tau$ ≤ T, and initiate the denoising process from that point. This design allows for more effective refinement and generation, leveraging the strengths of both LLM and diffusion models.
> >
> > - **Parallel Training:** We did not adopt the sequential training paradigm of FlowLLM, where a large language model (LLaMA-2) is first fine-tuned on the training dataset, and the samples generated from this LLM are then used to train the flow matching framework. Instead, in our approach, both the LLM and the diffusion module are trained in parallel using the same training dataset.
> >
> > - **Experimental Evaluation:** Finally, our experimental results demonstrate that CrysLLMGen outperforms FlowLLM across most standard benchmark metrics for the de novo material generation task (as shown in Table 3 of the paper) across different datasets. These results highlight the overall effectiveness of our proposed approach.
> >
> > [1] Miller, Benjamin Kurt, et al. "Flowmm: Generating materials with riemannian flow matching, ICML-2024
>
> - ***W2. Clarification on Architectural Differences and Performance Gains in CrysLLMGen***
>
> > We thank the reviewer for this insightful question. First, we clarify that both CrysLLMGen and FlowLLM use the same underlying LLM architecture (LLaMA-2) in the first stage of their respective hybrid pipelines. The key differences, and our primary innovations, lie in the second stage of the pipeline. Specifically, there are two main distinctions:
> >
> > - **Generative Module:** FlowLLM employs a flow-matching model as its generative component, whereas CrysLLMGen uses a diffusion model. We evaluated the performance of both flow-matching and diffusion models in the context of unconditional material generation and found that diffusion models consistently outperform flow-matching models across the majority of the evaluation metrics.
> >
> >
> >  - **Integration Strategy:** In FlowLLM, the material structures generated by the LLM are directly passed into the flow-matching module for refinement. In contrast, since LLM generated material structure are intermediate representations, we inject these representations into our diffusion model at an intermediate timestep τ, where 0 ≤ τ ≤ T, and initiate the denoising process from that point. This design allows for more effective refinement and generation, leveraging the strengths of both LLM and diffusion models.
> >
> >
> > These two innovations collectively contribute to the superior performance of CrysLLMGen compared to FlowLLM, as shown in Table 3.
>
> - ***W3. Ablation Study***
>
> > As suggested by the reviewer, we conducted an ablation study in which we replaced the FlowMatching module of FlowLLM with a diffusion model (following the DiffCSP framework). In specific, we first used data sampled from LLaMA-2 to train the diffusion model. During sampling, given a prompt, we first generate A, X, L from the LLM, which are then passed to the diffusion model (injecting at the final step T) for further refinement. The results we observed from this ablation are as follows:
> >
> > |Models|Struct Validity|Comp Validity|COV-Precision|COV-Recall|Density|#Elements|
> > |-|-|-|-|-|-|-|
> > |FlowLLM w\ Diffusion|99.88|82.50|98.54|97.55|0.724|0.5388|
> > |CrysLLMGen|99.94|93.78|99.84|98.52|0.664|0.459|
> >
> > We observe performance degradation. In specific, 12.03% performance degradation in compositional validity. These results suggest that training the diffusion model on true ground-truth data (rather than on LLM-sampled data), along with employing intermediate denoising to refine LLM-generated structures, enhances the effectiveness of our proposed hybrid framework.
>
>
> ***Questions:***
>
> - ***Q1. Clarification on #Element Metric Discrepancy***
>
> > We thank the reviewer for pointing this out. For consistency, we have reported benchmark results as presented in the respective original papers. The results for LLaMA-2 (7B) are taken from the Crystal-Text-LLM paper [2]. However, we also conducted our own sampling using the fine-tuned LLaMA-2 (7B) models, and the results we obtained for the de novo generation task are as follows:
> >
> > |Models|Struct Validity|Comp Validity|#Elements|
> > |-|-|-|-|
> > |LLaMA-2 7B|96.40|93.78|0.462|
> > |CrysLLMGen|99.94|93.78|0.459|
> >
> > We observe both compositional validity and # Element metrics are similar the two models. We will update it in the revised manuscript.
> >
> > [2] Gruver, Nate, et al. "Fine-tuned language models generate stable inorganic materials as text." ICLR 2024
>
> - ***Q2. Visualizations of intermediate and final samples***
>
> > We thank the reviewer for this valuable suggestion. We will include visualizations of a few generated material samples in the revised manuscript
>
> ***Paper Formatting Concerns***
>
> > We thank the reviewer for these thoughtful suggestions and appreciate the attention to detail regarding formatting and presentation.
> > - We will include representative examples of sampled structures in the revised manuscript, enabling experts to visually assess the quality of the generated materials.
> > - Regarding the Denoising Network section (Section 4.4), we agree that the technical depth may be better suited for the Supplementary Material. We will move this section accordingly to maintain the flow and readability of the main paper.
> > - We acknowledge the issue with Figure 1 and will replot it at higher resolution to ensure clarity and better visual quality.
> > - Additionally, we appreciate the recommendation to include examples of intermediate textual representations and details on coordinate tokenization. We will incorporate these elements in the revised manuscript or the camera-ready version to enhance reproducibility of our approach.

---

> > ### Comment · Reviewer_JFzg · 2025-08-05
> > **Concerns Regarding Rebuttal Focus and Evaluation Alignment**
> >
> > - **Reviewer’s Question Was About Novelty Relative to FlowLLM**
> >   The reviewer explicitly questioned whether the proposed architecture provides a meaningful conceptual advancement over FlowLLM, asking for clarity on what drives performance gains—architectural differences or model choices—particularly within the proposed *conditional generation* setting.
> >
> > - **Authors Responded with Unconditional Generation Results**
> >   The authors’ rebuttal and ablation studies focus primarily on *unconditional generation* tasks (e.g., comparing DiffCSP vs. FlowMM, and FlowLLM + Diffusion vs. CrysLLMGen). While these experiments may highlight performance differences between generative modules in general, they do not address the core motivation or novelty of CrysLLMGen in the *conditional generation* context.
> >
> > - **CrysLLMGen Is Framed as a Conditional Generation Method**
> >   The paper emphasizes capabilities such as text-conditioned or composition-conditioned generation as central contributions. Therefore, the authors' justification for choosing diffusion and the claims of superiority should be grounded in performance on *conditional tasks*.
> >
> > - **No Evidence That CrysLLMGen Offers Better Conditional Control**
> >   The rebuttal lacks experiments that demonstrate superior controllability, prompt fidelity, or robustness under diverse conditional prompts when compared to FlowLLM or other baselines. This gap undermines the stated motivation for combining LLMs and diffusion models.
> >
> > - **Misalignment in Comparison Setup**
> >   The ablation study compares CrysLLMGen with a variant of FlowLLM using a diffusion module. However, the two pipelines differ not only in architecture but also in their training data:
> >   *FlowLLM trains its flow model on LLM-generated samples*, while *CrysLLMGen trains its diffusion model on ground-truth crystal structures*.
> >   These differences confound the interpretation of performance gains and do not provide a clean architectural comparison.
> >
> > - **Actionable Recommendation**
> >   To substantiate their claims, the authors should include an ablation or comparison under the same *conditional generation* setting, using FlowLLM and CrysLLMGen with aligned prompts and consistent training data. This would help isolate the true impact of the proposed design. Without this, the current evaluation does not convincingly support the claimed advantages of the method.

---

### Official Review · Reviewer_A8AZ · 2025-07-03

**Clarity:** 3
**Significance:** 2
**Originality:** 2
**Rating:** 3
**Confidence:** 4

**Summary:**

The paper introduces a hybrid approach for crystal generation. Specifically, it first trains a large language model (LLM) to provide an initial structure, represented by A (discrete atomic types), F (fractional coordinates), and L (lattice matrix). This initial structure is then refined using an equivariant diffusion model. In this way, the approach combines the strengths of autoregressive (AR) models for generating atomic compositions and diffusion models for structure generation. The shown benchmarks of Crystal Structure Prediction and Denovo generation has shown the flexibility of the proposed framework.

**Questions:**

1. Could the authors provide a more in-depth discussion of the performance gap between Llama-2 (7B) and CrysLLMGen (7B) to better illustrate the effectiveness of their approach?

2. I am also curious about the choice of using diffusion in the second phase—are there specific benefits or motivations for selecting this method?

**Ethical Concerns:**

["NO or VERY MINOR ethics concerns only"]

**Final Justification:**

empirical strong paper with limited novelty

**Limitations:**

yes

**Quality:**

3

**Strengths And Weaknesses:**

**Strengths**

1. Leveraging the generative capabilities of LLMs for composition generation, combined with fine-tuning via a domain- or geometry-specific model, is a simple yet effective idea. Bridging advanced AI tools with LLMs is an important and promising direction that should be encouraged.
2. I appreciate the effort in integrating this pipeline to enhance performance across multiple benchmarks. The empirical results and the text-based generation capabilities demonstrated are impressive.
The paper is well written, with sufficient introduction and technical details to enable a full understanding of both the method and the results.

**Weaknesses**

1. Although the paper includes a paragraph comparing the proposed approach to FlowLLM, I remain unconvinced by the differences highlighted. The key motivation for FlowLLM's use of sampled data during training is to maintain alignment in the inference process under distribution shift [1]. The minor difference of fixing atom types in the second phase does not, in my view, constitute a significant distinction in design. This raises major concerns regarding the technical novelty and contribution of the proposed approach.

2. Upon closer examination of the results, the base 7B LLM alone already achieves very strong overall performance, making the incremental improvement from the diffusion process seem marginal. I recommend the authors provide a more detailed analysis of the inference efficiency in the two-phase approach, and further clarify the motivation for the framework.

3. The paper lacks comparison with some important baselines and citations of related works, including [2,3,4]. Comparisons with state-of-the-art or advanced models are essential.

[1] FlowLLM: Flow Matching for Material Generation with Large Language Models as Base Distributions

[2] MatterGen: a generative model for inorganic materials design

[3] A Periodic Bayesian Flow for Material Generation

[4] SymmCD: Symmetry-Preserving Crystal Generation with Diffusion Models

---

> ### Author Rebuttal · Authors · 2025-07-30
>
> We thank the reviewer for the valuable feedback. Below, we provide detailed point-by-point responses to each of the comments and questions:
>
> ***Weaknesses :***
>
>  - ***W1. Difference with FlowLLM***
>
> > By design, our proposed CrysLLMGen bears some resemblance to FlowLLM; however, it is not merely an adaptation of FlowLLM’s pipeline with the flow-matching module replaced by a diffusion model. The key innovations and novel aspects of CrysLLMGen are as follows:
> >
> > - **Generative Module:** FlowLLM employs a flow-matching model as its generative component, whereas CrysLLMGen uses a diffusion model. We analyzed the performance of both the flow matching and diffusion models in the context of unconditional material generation. Our observations indicate that DiffCSP, the diffusion-based model for material generation, outperforms FlowMM, the flow matching model, across most of the evaluation metrics for this task. Referencing the FlowMM paper [1], the summary of results from de novo generation on the MP-20 dataset is as follows:
> >
> > |Models|Struct Validity|Comp Validity|COV-Precision|COV-Recall|Density|#Elements|Stability Rate|SUN Rate|
> > |-|-|-|-|-|-|-|-|-|
> > |DiffCSP(1000 Steps)|**100**|**83.25**|**99.76**|**99.71**|0.35|0.125|**5.06**|**3.34**|
> > |FLowMM(1000 Steps)|96.85|83.19|99.58|99.49|**0.239**|**0.083**|4.65|2.34|
> >
> > These observations indicate that the diffusion model is a more natural choice for material generation compared to flow matching, as it consistently produces more valid and stable materials.
> >
> > - **Integration Strategy:** In FlowLLM, the material structures generated by the LLM are directly passed into the flow-matching module for refinement. In contrast, since LLM generated material structure are intermediate representations, we inject these representations into our diffusion model at an intermediate timestep $\tau$ , where 0 ≤ $\tau$ ≤ T, and initiate the denoising process from that point. This design allows for more effective refinement and generation, leveraging the strengths of both LLM and diffusion models.
> >
> > - **Parallel Training:** We did not adopt the sequential training paradigm of FlowLLM, where a large language model (LLaMA-2) is first fine-tuned on the training dataset, and the samples generated from this LLM are then used to train the flow matching framework. Instead, in our approach, both the LLM and the diffusion module are trained in parallel using the same training dataset.
> >
> > - **Experimental Evaluation:** Finally, our experimental results demonstrate that CrysLLMGen outperforms FlowLLM across most standard benchmark metrics for the de novo material generation task (as shown in Table 3 of the paper) across different datasets. These results highlight the overall effectiveness of our proposed approach.
> >
> > [1] Miller, Benjamin Kurt, et al. Flowmm: Generating materials with Riemannian flow matching, ICML-2024
>
> - ***W2. Improvement of CrysLLMGen over LLMs***
>
> >The reviewer’s observation regarding the incremental improvement of CrysLLMGen over the base 7B LLM model may have come from the results of the de novo generation task on the Perov dataset (a simpler benchmark) reported in Table 3. However, if one examine the results on the more challenging MP-20 benchmark dataset, there are significant gains. Here is the summary of the results for the de novo generation task and stability metrics (detailed results are provided in Tables 3 and 4 of the paper):
> >
> > |Models|Struct Validity|Comp Validity|COV-Precision|COV-Recall|Density|#Elements|Meta Stability Rate| MSUN Rate | Stability Rate|SUN Rate|
> > |-|-|-|-|-|-|-|-|-|-|-|
> > |LLaMA-2 7B      |96.40|93.30|94.90|91.10|3.610|1.060|36.27|23.92|6.910|5.29|
> > |CrysLLMGen      |99.94|93.78|99.84|98.52|0.664|0.459|45.20|35.94|11.98|9.21|
> > |% Improvements  |3.672|0.514|5.205|8.144|81.60|56.69|24.62|50.25|73.37|74.10|
> >
> > We clearly observe that CrysLLMGen outperforms the LLaMA baseline across most benchmark metrics by a significant margin. Notably, the overall average improvement across all metrics is 37.81%.
> >
> > We observe that CrysLLMGen significantly outperforms the LLaMA-2 7B model on most of the metrics for material generation.
>
>
> - ***W3. Comparing with additional generative models: Mattergen, SymmCD, CrysBFN***
>
> > We thank the reviewer for highlighting the baseline models. All the mentioned baselines have been evaluated on the De Novo Generation Task using the MP-20 dataset, and the results are as follows:
> >
> > |Models|Struct Validity|Comp Validity|COV-Precision|COV-Recall|Density|#Elements|
> > |-|-|-|-|-|-|-|
> > *Already reported*
> > |DiffCSP    |100    |83.25|99.76|99.71|0.3502|0.3398|
> > |FlowMM     |96.85  |83.19|99.58|99.49|0.2390|0.083|
> > *New experiments*
> > |Mattergen  |100    |86.34|99.45|99.59|0.4597|0.2538|
> > |SymmCD     |88.10  |85.32|96.82|99.53|0.201|0.3639|
> > |CrysBFN    |100    |87.51|99.79|99.09|0.206|0.1628|
> >
> > Overall, the results follow similar trends as observed with other generative models reported in Table 3. Similar to other diffusion- and flow-based generative models, these models generally perform well in structural validity but struggle with discrete components, such as accurately identifying atomic types, leading to lower compositional validity. While there may be some quantitative differences, the overall observations remain consistent across most of these models. We will include these results in the revised manuscript and appropriately cite these works.
>
>
> ***Questions:***
>
> - ***Q1. In-depth discussion of the performance gap between Llama-2 (7B) and CrysLLMGen (7B)***
>
> > Check W2.
>
> - ***Q2. Motivation behind Choosing Diffusion Model***
>
> > In an unconditional setup, DiffCSP performs better than FlowMM. We analyzed the performance of both the flow matching and diffusion models in the context of unconditional material generation. Our observations indicate that DiffCSP, the diffusion-based model for material generation, outperforms FlowMM, the flow matching model, across most benchmark metrics (6 out of 8) for this task. Referencing from the FlowMM paper, the summary of results from de novo generation on the MP-20 dataset is as follows:
> >
> > |Models|Struct Validity|Comp Validity|COV-Precision|COV-Recall|Density|#Elements|Stability Rate|SUN Rate|
> > |-|-|-|-|-|-|-|-|-|
> > |DiffCSP(1000 Steps)|**100**|**83.25**|**99.76**|**99.71**|0.350|0.125|**5.06**|**3.34**|
> > |FLowMM(1000 Steps)|96.85|83.19|99.58|99.49|**0.239**|**0.083**|4.65|2.34|
> >
> >
> > These observations indicate that the diffusion model is a more natural choice for material generation compared to flow matching, as it consistently produces more valid and stable materials.

---

> > ### Author Response · Authors · 2025-08-05
> > **Looking forward to your feedback**
> >
> > Dear Reviewer,
> >
> > Thank you for your valuable feedback and constructive comments.
> >
> > In our rebuttal, we have provided additional experiments and enhanced explanations, and we hope we have addressed all the concerns raised by you. We are open to further discussions and are happy to clarify any remaining doubts.
> >
> > If there are still any outstanding issues, we kindly request you to share them with us. Otherwise, we would greatly appreciate it if you could consider revising the score.
> > We look forward to your response.
> >
> > Thank you,
> >
> > The Authors

---

> > > ### Comment · Reviewer_A8AZ · 2025-08-07
> > > **Response to Rebuttal**
> > >
> > > Thank you for the detailed rebuttal and for providing additional experiments. These have helped clarify several aspects of the proposed framework.
> > >
> > > My updated assessment is as follows:
> > >
> > > 1. On the Comparison with FlowLLM and Component Ablation
> > >
> > > I appreciate the detailed comparison with FlowLLM in the rebuttal, which highlights several key differences. However, the specific contributions of each of your three main design choices remain somewhat unclear to me.
> > >
> > > To better understand your work, could you please provide a more direct clarification on the following:
> > >
> > > Which specific components are most responsible for the superior performance over FlowLLM?
> > > How do these three designs individually contribute to the overall framework's success?
> > >
> > > This is particularly important given your claim that diffusion models inherently hold an advantage over flow-matching approaches in this context. A clear ablation or analysis here would be very compelling.
> > >
> > > 2. On the Related Works and Baselines
> > >
> > > I want to thank you for including the additional baselines. This was very helpful and has strengthened my understanding of the necessity of handling discrete variables in LLMs.
> > >
> > > I would raise my score upon receiving further clarification regarding the individual effects of your proposed components.

---

> > > > ### Author Response · Authors · 2025-08-07
> > > >
> > > > Dear Reviewer A8AZ,
> > > >
> > > > We thank you for the positive feedback. Below we will clarify the concern you raise.
> > > >
> > > > ***Motivation for design choices of CrysLLMGen:***
> > > >
> > > > We hypothesize that the combination of all three key factors significantly contributed to the observed improvements. Let us begin by outlining the motivation behind our proposed framework.
> > > >
> > > > Our goal was to harness the strengths of both large language models (LLMs) and diffusion-based generative models to generate more stable and novel material structures. We identified FlowLLM as a recent state-of-the-art approach. Notably, both FlowLLM and our Crystal-Text-LLM adopt the LLaMA-2 model as the LLM backbone, and we maintained this choice to enable a fair comparison.
> > > >
> > > > Next, we examined whether flow matching or diffusion is a more effective generative paradigm for material generation. And as mentioned above, from the results reported in the FlowMM paper [1], diffusion model is a more natural choice for material generation compared to flow matching, as it consistently produces more valid and stable materials.
> > > >
> > > > We also identified a critical limitation in FlowLLM’s training setup: its flow matching component is trained not on the true (gold) data, but on LLM-generated outputs. This introduces a distributional mismatch—the model learns to refine structures based on the estimated distribution rather than the true data distribution, which can degrade performance.
> > > >
> > > > In our approach, we address this by training the diffusion model directly on the gold data. However, we observed that the LLM-generated structures are not simple noisy inputs, as expected at the beginning of the diffusion process, but instead resemble intermediate structures. To accommodate this, during inference we inject the LLM-generated structures at an intermediate diffusion step τ\tau (selected based on validation performance), allowing the diffusion model to further refine them toward realistic material structures.
> > > >
> > > >
> > > > ***Ablation Studies***
> > > >
> > > > However, to enable a more thorough assessment, we conducted a couple of ablation studies.
> > > >
> > > > **Regarding Training Data:**
> > > >
> > > > First, we replaced the FlowMatching module in FlowLLM with a diffusion model and trained this diffusion model using the same LLM, generated data as used in FlowLLM. We refer to this variant as FlowLLM w/ Diffusion. The performance results observed for the MP-20 dataset are as follows:
> > > >
> > > > |Models|Struct Validity|Comp Validity|COV-Precision|COV-Recall|Density|#Elements|
> > > > |-|-|-|-|-|-|-|
> > > > |FlowLLM w\ Diffusion |99.88|82.50|98.54|97.55|0.724|0.539|
> > > > |CrysLLMGen           |99.94|93.78|99.84|98.52|0.664|0.459|
> > > >
> > > > We observe a clear performance degradation in this setting, ***specifically, a 12.03% drop in compositional validity***. These results highlight the importance of training the diffusion model on true ground-truth data. This choice significantly enhances the effectiveness of our proposed hybrid framework.
> > > >
> > > >
> > > > **Regarding Integration Steps:**
> > > >
> > > > Next, we performed an ablation study to probe our proposed CrysLLMGen architecture by modifying the point of integration of the LLM-generated structure. Specifically, during sampling, we injected the LLM-generated structure at the final step (T-th step) and performed the full T-step denoising process from there.
> > > >
> > > > The results for the MP-20 dataset under this setup are as follows:
> > > >
> > > > |Models|Struct Validity|Comp Validity|COV-Precision|COV-Recall|Density|#Elements|
> > > > |-|-|-|-|-|-|-|
> > > > |CrysLLMGen (integrate at T step)     |99.21|93.78|81.29|93.93|5.420|0.650|
> > > > |CrysLLMGen (integrate at \tau step)  |99.94|93.78|99.84|98.52|0.664|0.459|
> > > >
> > > > We observe performance degradation, notably for Coverage and Property statistics. These results shows impact of integration strategy.
> > > >
> > > > We will include these ablation results in the revised manuscript.

---

> > > > > ### Author Response · Authors · 2025-08-08
> > > > > **Looking forward to your feedback**
> > > > >
> > > > > Dear Reviewer A8AZ,
> > > > >
> > > > > Thank you once again for your thoughtful and constructive feedback.
> > > > >
> > > > > We have provided detailed responses explaining the motivation behind the design choices of CrysLLMGen, along with ablation results demonstrating the impact of its individual components. We hope these additional results and analyses adequately address all the concerns you raised.
> > > > >
> > > > > If there are any remaining issues or if further clarification is needed, please let us know—we would be happy to address them. Otherwise, we kindly request that you consider revising your score in light of these updates.
> > > > >
> > > > > We look forward to your response.
> > > > >
> > > > > Thank you,
> > > > >
> > > > > The Authors

---

> > > > > > ### Comment · Reviewer_A8AZ · 2025-08-08
> > > > > > **Response to additional results**
> > > > > >
> > > > > > Thank you for providing the new experiments. While I still have concerns about the technical novelty, I have raised my score to reflect the strengthened empirical results.

---

> > ### Comment · Reviewer_JFzg · 2025-08-05
> > **Clarification Needed on Novelty, Efficiency, Conditioning, and Evaluation Transparency**
> >
> > I appreciate the detailed review and would like to echo several of the concerns raised, particularly regarding the technical novelty, practical value, and evaluation transparency of the proposed approach. Below are several points that I believe still require clarification:
> >
> > Lack of Clear Conceptual Novelty
> > The stated design difference—injecting LLM-generated structures at intermediate diffusion steps and training modules in parallel—seems more like an implementation detail than a conceptual or modeling innovation. This design choice is not accompanied by theoretical justification or substantial algorithmic reformulation. For NeurIPS standards, a clearer explanation is needed to demonstrate why this integration constitutes a novel modeling contribution rather than a reconfiguration of existing components.
> >
> > Efficiency Trade-off Not Discussed
> > The rebuttal highlights moderate improvements over the LLaMA-2 baseline, but does not provide a thorough analysis of computational cost, latency, or scalability of the two-stage pipeline. Given the added complexity of combining LLM and diffusion models, a discussion on efficiency vs. performance gain is necessary. Without it, it's difficult to assess the practical value of the proposed hybrid method.
> >
> > Conditioning and Prompt Limitations
> > The method relies on natural language prompts to control composition and structure. However, structured prompts are known to be brittle, dataset-dependent, and lack generalizability. The current pipeline does not introduce any mechanism for more robust conditioning or controllability during generation. This limitation weakens the applicability of the model to open-ended or high-stakes materials discovery tasks where precision and control are crucial.
> >
> > Unclear Handling of Invalid LLM Outputs
> > The paper still does not clarify whether invalid LLM generations (e.g., compositions that cannot be parsed or violate physical constraints) are filtered before or during evaluation. It also remains unclear whether these are:
> >
> > counted as failures (which would lower the reported performance), or
> >
> > silently discarded (which could artificially inflate scores).
> > This is a crucial point, especially when reporting high validity and stability metrics, and should be transparently documented.
> >
> > New Experiments Do Not Clearly Justify Importance
> > In the newly added baselines, the proposed method does not outperform MatterGen on several critical metrics, including:
> >
> > Structural Validity
> >
> > COV-Recall
> >
> > Density
> >
> > Number of Elements
> >
> > This raises further questions about the motivational strength of the proposed framework. If existing single-stage models like MatterGen match or surpass the performance of the hybrid method, the practical justification and contribution of the new pipeline remain unclear. A more thorough discussion comparing these results — and why one might still prefer CrysLLMGen — should be included in the main paper.

---

### Decision · Program_Chairs · 2025-09-17

**Decision:**

Accept (poster)

**Comment:**

The paper proposes a hybrid LLM–diffusion model for crystal material generation that demonstrates competitive empirical performance. Reviewers agreed the paper is clearly written and that the empirical evaluation is strong, though concerns were raised about limited novelty, lack of joint training between the LLM and diffusion model, evaluation transparency, and missing baselines. The authors addressed most of these convincingly during rebuttal, and with three of four reviewers recommending acceptance, I recommend acceptance as well.